# Aggregation Buffer: Revisiting DropEdge with a New Parameter Block

**Dooho Lee** [1]   **Myeong Kong** [1]   **Sagad Hamid** [1 2]   **Cheonwoo Lee** [1]   **Jaemin Yoo** [1]

## Abstract

We revisit DropEdge, a data augmentation technique for GNNs which randomly removes edges to expose diverse graph structures during training. While being a promising approach to effectively reduce overfitting on specific connections in the graph, we observe that its potential performance gain in supervised learning tasks is significantly limited. To understand why, we provide a theoretical analysis showing that the limited performance of DropEdge comes from the fundamental limitation that exists in many GNN architectures. Based on this analysis, we propose *Aggregation Buffer*, a parameter block specifically designed to improve the robustness of GNNs by addressing the limitation of DropEdge. Our method is compatible with any GNN model, and shows consistent performance improvements on multiple datasets. Moreover, our method effectively addresses well-known problems such as degree bias or structural disparity as a unifying solution. Code and datasets are available at https://github.com/dooho00/agg-buffer.

## 1. Introduction

Graph-structured data are pervasive across various research fields and real-world applications, as graphs naturally capture essential relationships among entities in complex systems. Graph neural networks (GNNs) have emerged as a powerful framework to effectively incorporate these relationships for graph-related tasks. In contrast to traditional multi-layer perceptrons (MLPs), which solely consider node features, GNNs additionally take advantage of edge information to incorporate crucial interrelations between node features (Kipf & Welling, 2017; Gasteiger et al., 2019; Hamilton et al., 2017; Veličković et al., 2018). As a consequence, GNNs are able to account for interaction patterns and structural dependencies, a source of knowledge that enables improving the performance in semi-supervised learning tasks, even with limited observations (Ying et al., 2018; Brody et al., 2022; Song et al., 2022).

While leveraging edge structure has proven highly effective, it often makes a GNN overfit to certain structural properties of nodes mainly observed in the training data. As a result, the model's performance suffers considerably in the presence of *structural inconsistencies*. For example, it is widely known that GNNs perform worse on low-degree nodes than on high-degree nodes even when their features are highly informative, since high-degree nodes are the main source of information for their training (Tang et al., 2020; Liu et al., 2023; Subramonian et al., 2024). Moreover, GNNs exhibit poor accuracy on nodes whose neighbors have conflicting structural properties, such as heterophilous neighbors in homophilous graphs, or vice versa (Wang et al., 2024; Mao et al., 2024). These problems clearly highlight the two faces of GNNs–their reliance on edge structure is the key to their success, while also making them more vulnerable.

Common approaches to enhance robustness against input data variations in supervised learning are *random dropping* techniques such as DropOut (Srivastava et al., 2014). For GNNs, DropEdge (Rong et al., 2020) has been introduced as a means to increase the robustness against edge perturbations. DropEdge removes a random subset of edges at each iteration, exposing a GNN to diverse structural information. However, the performance gain by DropEdge is limited in practice, and DropEdge is typically excluded from the standard hyperparameter search space of GNNs in benchmark studies (Dwivedi et al., 2023; Luo et al., 2024).

In this work, we provide a theoretical analysis on the reason why DropEdge fails. We study the *objective shift* caused by DropEdge and highlight the implicit bias-robustness tradeoff in its objective function. Then, we prove that the failure of DropEdge is not because of its algorithm, but the inductive bias existing in most GNN architectures, based on the concept of *discrepancy bound* in comparison to MLPs.

Building on these insights, we propose *Aggregation Buffer* ($AGG_B$), a new parameter block which can be integrated to to any trained GNN as a post-processing procedure. $AGG_B$ effectively addresses the architectural limitation of GNNs,

[1]School of Electrical Engineering, KAIST, Daejeon, Republic of Korea [2]Computer Science Department, University of Münster, Münster, Germany. Correspondence to: Jaemin Yoo <jaemin@kaist.ac.kr>.

*Proceedings of the $42^{nd}$ International Conference on Machine Learning*, Vancouver, Canada, PMLR 267, 2025. Copyright 2025 by the author(s).

allowing DropEdge to significantly enhance the robustness of GNNs compared to its original working mechanism. We demonstrate the effectiveness of $\text{AGG}_B$ in improving the robustness and overall accuracy of GNNs across 12 node classification benchmarks. In addition, we show that $\text{AGG}_B$ works as a unifying solution to structural inconsistencies such as degree bias and structural disparity, both of which arise from structural variations in graph datasets.

## 2. Preliminaries

**Notation.** Let $\mathcal{G} = (V, E)$ be an undirected graph, where $V$ is the set of nodes and $E$ is the set of edges. We denote the adjacency matrix by $\boldsymbol{A} \in \{0, 1\}^{|V| \times |V|}$, where $a_{ij} = 1$ if there is an edge between nodes $i$ and $j$, and $a_{ij} = 0$ otherwise. The node feature matrix is denoted by $\boldsymbol{X} \in \mathbb{R}^{|V| \times d_0}$, where $d_0$ is the dimensionality of features.

**Graph Neural Network.** A graph neural network (GNN) consists of multiple layers, each performing two key operations: *aggregate* (AGG) and *update* (UPDATE) (Gilmer et al., 2017; Hu et al., 2020b). For each node, AGG gathers information from its neighboring nodes in the graph structure, while UPDATE combines the aggregated information with the node's previous representation. With $\boldsymbol{H}^{(0)} = \boldsymbol{X}$, we formally define the $l$-th layer $\boldsymbol{H}^{(l)} \in \mathbb{R}^{|V| \times d_l}$ as

$$\boldsymbol{H}_{\mathcal{N}}^{(l)} = \text{AGG}^{(l)}(\boldsymbol{H}^{(l-1)}, \boldsymbol{A}),$$
$$\boldsymbol{H}^{(l)} = \text{UPDATE}^{(l)}(\boldsymbol{H}_{\mathcal{N}}^{(l)}, \boldsymbol{H}^{(l-1)}),$$

where $d_l$ is the dimensionality of embeddings from the $l$-th layer, and a learnable weight matrix $\boldsymbol{W}^{(l)} \in \mathbb{R}^{d_{l-1} \times d_l}$ is typically used to transform representations between layers. We also denote $\boldsymbol{H}^{(s:t)} \in \mathbb{R}^{|V| \times (d_s + \cdots + d_t)}$ as the concatenation of node embeddings from layer $s$ to $t$ along the feature dimension, as $\boldsymbol{H}^{(s:t)} = \boldsymbol{H}^{(s)} \| \ldots \| \boldsymbol{H}^{(t)}$, where $\|$ denotes the concatenation operator and $s < t$.

## 3. Revisiting DropEdge

We give an overview of DropEdge and formalize its *objective shift* in node-level tasks. We then analyze the implicit bias-robustness trade-off in its objective and exhibit an unexpected failure of DropEdge on improving robustness.

### 3.1. Overview of DropEdge

DropEdge (Rong et al., 2020) is a data augmentation algorithm for improving the generalizability of GNNs by introducing stochasticity during its training. More specifically, it modifies the graph's adjacency matrix $\boldsymbol{A}$ using a binary mask $\boldsymbol{M} \in \{0, 1\}^{|V| \times |V|}$, by creating an adjacency matrix $\tilde{\boldsymbol{A}} = \boldsymbol{M} \odot \boldsymbol{A}$, where $\odot$ is the element-wise multiplication between matrices. The matrix $\boldsymbol{M}$ is generated randomly to drop a subset of edges by setting their values to zero.

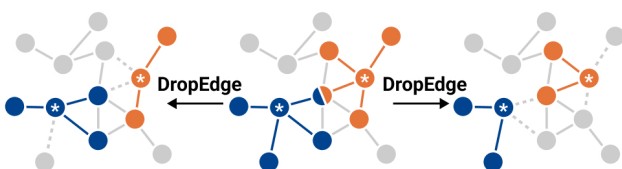

*Figure 1.* DropEdge generates various reduced rooted subgraphs for center nodes (*) by randomly removing edges.

In node-level tasks, a GNN can be considered as taking the $k$-hop subgraph of each node as its input. For each node $i$, the edge removal operation in DropEdge can be interpreted as transforming the rooted subgraph $\mathcal{G}_i$, centered on node $i$, into a reduced rooted subgraph, denoted as $\tilde{\mathcal{G}}_i$. Figure 1 illustrates how DropEdge modifies a rooted subgraph.

**Definition 3.1** (Rooted Subgraph). A *rooted subgraph* $\mathcal{G}_i = (V_i, E_i)$ is a $k$-hop subgraph centered on node $i$, where $V_i$ is the set of nodes within the $k$-hop neighborhood of node $i$ and $E_i$ denotes the set of edges between nodes in $V_i$.

**Definition 3.2** (Reduced Rooted Subgraph). Given a rooted subgraph $\mathcal{G}_i$ as an input, a *reduced rooted subgraph* $\tilde{\mathcal{G}}_i = (\tilde{V}_i, \tilde{E}_i)$ is a subgraph of $\mathcal{G}_i$ created by DropEdge, where $\tilde{V}_i \subseteq V_i$ and $\tilde{E}_i \subseteq E_i$ is the edge set induced by $\tilde{V}_i$.

### 3.2. Bias-Robustness Trade-off

In a typical classification task, the objective is to minimize the Kullback-Leibler (KL) divergence between the true posterior $P(\boldsymbol{y}_i | \mathcal{G}_i)$ and the modeled one $Q(\boldsymbol{y}_i | \mathcal{G}_i)$ as

$$\mathcal{L}(\theta) = D_{\text{KL}}(P(\boldsymbol{y}_i | \mathcal{G}_i) \| Q(\boldsymbol{y}_i | \mathcal{G}_i)), \quad (1)$$

where $\theta$ is the set of parameters used for modeling $Q$. When DropEdge is used during the training of a GNN, it perturbs the given rooted subgraph and creates a reduced subgraph $\tilde{\mathcal{G}}_i$ which leads to the following shifted objective function:

$$\tilde{\mathcal{L}}(\theta) = D_{\text{KL}}(P(\boldsymbol{y}_i | \mathcal{G}_i) \| Q(\boldsymbol{y}_i | \tilde{\mathcal{G}}_i)). \quad (2)$$

The shifted objective $\tilde{\mathcal{L}}$ can be decomposed as follows:

$$\tilde{\mathcal{L}}(\theta) = D_{\text{KL}}(P(\boldsymbol{y}_i | \mathcal{G}_i) \| Q(\boldsymbol{y}_i | \mathcal{G}_i)) +$$
$$\mathbb{E}_P[\log Q(\boldsymbol{y}_i | \mathcal{G}_i) - \log Q(\boldsymbol{y}_i | \tilde{\mathcal{G}}_i)]. \quad (3)$$

The first term corresponds to the standard objective function $\mathcal{L}(\theta)$ in Equation 1, which ensures that optimizing $\tilde{\mathcal{L}}(\theta)$ remains aligned with its intended purpose. It particularly measures how well the model approximates the true posterior and can be referred to as *bias*, as it relies on observed labels collected to represent the true distribution.

The second term measures the expected difference between $\log Q(\boldsymbol{y}_i | \mathcal{G}_i)$ and $\log Q(\boldsymbol{y}_i | \tilde{\mathcal{G}}_i)$. This can be understood as measuring the *robustness* of a GNN against edge perturbations, as it is minimized s.t. the GNN produces consistent

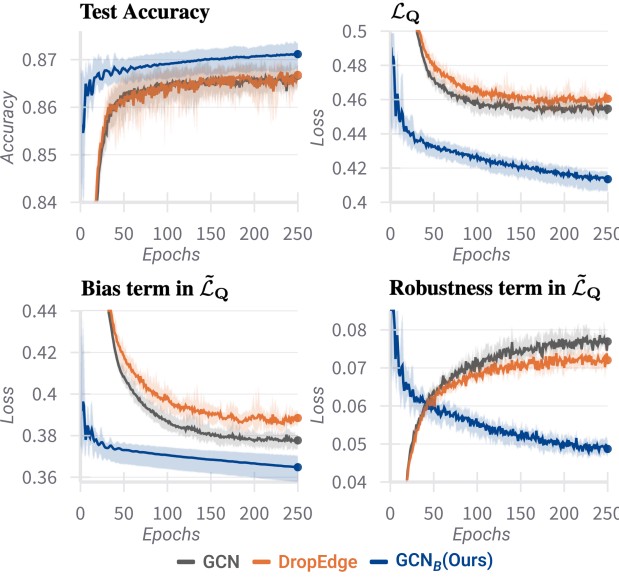

**Figure 2.** Accuracy and loss terms on test data during the training of a GCN at PubMed. We illustrate the average of 10 independent runs, with shaded regions representing the minimum and maximum values. While DropEdge decreases the robustness term compared to standard GNNs, it leads to increasing the bias term, eventually resulting in similar test accuracy to standard GNNs.

predictions for $\mathcal{G}_i$ and $\tilde{\mathcal{G}}_i$. However, as the expectation involves the true distribution $P$, it can only be computed if $P$ is known. By assuming that $Q$ is sufficiently close to $P$, we can rewrite $\tilde{\mathcal{L}}$ as follows:

$$\tilde{\mathcal{L}}_{\mathrm{Q}}(\theta) = D_{\mathrm{KL}}(P(\boldsymbol{y}_i|\mathcal{G}_i)\|Q(\boldsymbol{y}_i|\mathcal{G}_i)) + \\ D_{\mathrm{KL}}(Q(\boldsymbol{y}_i|\mathcal{G}_i)\|Q(\boldsymbol{y}_i|\tilde{\mathcal{G}}_i)). \quad (4)$$

We discuss the validity of this approximation in Appendix E. The interplay between the terms in $\tilde{\mathcal{L}}_{\mathrm{Q}}$ naturally introduces a *bias-robustness trade-off*. The first term, which is equal to $\mathcal{L}$, enables learning the true posterior accurately. The second term works as a regularizer, promoting consistency across different reduced rooted subgraphs. Finding an optimal balance between bias and robustness is key to maximize the performance of GNNs on unseen test graphs.

### 3.3. Unexpected Failure of DropEdge

DropEdge is designed to enhance the robustness of GNNs against structural perturbations. To evaluate its effectiveness, we train a GNN with and without DropEdge and measure $\tilde{\mathcal{L}}_{\mathrm{Q}}$ on the test set. As shown in Figure 2, DropEdge successfully regularizes the robustness term compared to standard GNNs. However, this comes at the cost of increasing the bias term, leading to a similar total $\tilde{\mathcal{L}}_{\mathrm{Q}}$ and no overall performance improvement. It is notable that we have carefully tuned the drop ratio of DropEdge for this example, suggest-

ing that other drop ratios would lead to degradation.

This behavior is consistently observed across all datasets in our experiments, raising the question of whether DropEdge can truly improve the performance. While a trade-off between bias and robustness is expected, this outcome is unusual compared to data augmentation methods in other domains (Srivastava et al., 2014; DeVries, 2017; Hou et al., 2022). In most cases, small perturbations of data do not significantly interfere with the primary learning objective, allowing robustness optimization to improve generalization. However, in GNNs trained with DropEdge, optimizing robustness immediately increases the bias term on test data, preventing sufficient robustness to be achieved.

This phenomenon highlights a fundamental challenge: the minimization of $\tilde{\mathcal{L}}_{\mathrm{Q}}$, in terms of both bias and robustness, is inherently difficult to achieve within the standard training framework of GNNs, limiting the effectiveness of DropEdge and similar techniques in improving edge-robustness.

### 3.4. Reason of the Failure: Core Limitations of GNNs

The robustness term in $\tilde{\mathcal{L}}_{\mathrm{Q}}$ can be optimized only when a GNN is able to produce similar outputs for different adjacency matrices, namely $\boldsymbol{A}$ and $\tilde{\boldsymbol{A}}$. To study the poor efficacy of DropEdge, we analyze how well a GNN can bound the difference between its outputs given different inputs, which we refer to as the *discrepancy bound*. Our key observation is that the failure of DropEdge is not rooted in its algorithm but rather in the inductive bias of GNNs, suggesting that it cannot be addressed optimally with existing GNN layers.

**Definition 3.3** (Discrepancy bound). Let $\boldsymbol{H}_1^{(l)}$ and $\boldsymbol{H}_2^{(l)}$ be the outputs of the $l$-th layer of a network $f$ given different inputs $\boldsymbol{H}_1^{(l-1)}$ and $\boldsymbol{H}_2^{(l-1)}$. The discrepancy bound of $f$ at the $l$-th layer is a constant $C$, such that

$$\|\boldsymbol{H}_1^{(l)} - \boldsymbol{H}_2^{(l)}\|_2 \leq C\|\boldsymbol{H}_1^{(l-1)} - \boldsymbol{H}_2^{(l-1)}\|_2,$$

where $C$ is independent of the specific inputs.

As a comparison, we first study the discrepancy bound of MLPs in Lemma 3.5 and move on to GNNs. Proofs for all theoretical results in this section are in Appendix C.

**Lemma 3.4.** *Commonly used activation functions—ReLU, Sigmoid, and GELU—and parameterized linear transformation satisfy Lipschitz continuity.*

**Lemma 3.5.** *Given an MLP with activation function $\sigma$, the discrepancy bound at the $l$-th layer is $C = L_\sigma\|\boldsymbol{W}^{(l)}\|_2$ where $L_\sigma$ is the Lipschitz constant of $\sigma$.*

**Theorem 3.6.** *In an $L$-layer MLP with activation function $\sigma$, the discrepancy bound at the $L$-th layer can be derived for every intermediate layer $l < L$ as*

$$\|\boldsymbol{H}_1^{(L)} - \boldsymbol{H}_2^{(L)}\|_2 \leq C\|\boldsymbol{H}_1^{(l)} - \boldsymbol{H}_2^{(l)}\|_2,$$

where $C = L_\sigma^{(L-l)} \prod_{i=l+1}^{L} \|\boldsymbol{W}^{(i)}\|_2$.

Theorem 3.6 implies that reducing discrepancies in intermediate representations can minimize discrepancies in the final output, allowing parameters in each layer to effectively contribute to the model's robustness. On the other hand, the linear discrepancy bound does not hold for GNNs. We formalize this observation in Theorem 3.8.

**Lemma 3.7.** *Commonly used aggregation functions in GNNs—regular, random walk normalized, and symmetric normalized—satisfy Lipschitz continuity.*

**Theorem 3.8.** *Given a graph convolutional network (GCN) with any non-linear activation function $\sigma$ and different adjacency matrices $\boldsymbol{A}_1$ and $\boldsymbol{A}_2$, the discrepancy bound cannot be established as a constant $C$ independent of the input.*

**Theorem 3.9.** *Under the same conditions as Theorem 3.8, the discrepancy of a GCN at layer $l$ is bounded as*

$$\|\boldsymbol{H}_1^{(l)} - \boldsymbol{H}_2^{(l)}\|_2 \le C_1 \|\boldsymbol{H}_1^{(l-1)} - \boldsymbol{H}_2^{(l-1)}\|_2 + C_2,$$

*where $C_1 = L_\sigma \|\boldsymbol{W}^{(l)}\|_2$, $C_2 = C_1|V|\|\hat{\boldsymbol{A}}_1 - \hat{\boldsymbol{A}}_2\|_2$, and $\hat{\boldsymbol{A}}$ is the normalized adjacency matrices of $\boldsymbol{A}$.*

The key difference between GNNs and MLPs arises from the AGG operation in GNN layers. While the inclusion of the AGG operation enables a GNN to utilize the graph structure, it becomes problematic when aiming for robustness under different adjacency matrices. As demonstrated in Theorem 3.9, discrepancies can arise purely due to differences in the adjacency matrices, as a form of $C_2$, even if the pre-aggregation representations are identical. This issue ultimately hinders the optimization of the robustness term in $\tilde{\mathcal{L}}_\mathrm{Q}$, as observed in Section 3.3.

# 4. Achieving Edge-Robustness

Our analysis in Section 3 shows the difficulty of optimizing $\tilde{\mathcal{L}}_\mathrm{Q}$ due to the nature of GNNs. As a solution, we propose *Aggregation Buffer* ($\mathrm{AGG}_B$), a new parameter block which can be integrated into a GNN's backbone as illustrated in Figure 3. $\mathrm{AGG}_B$ is specifically designed to refine the output of the AGG operation, mitigating discrepancies caused by variations in the graph structure introduced by DropEdge.

## 4.1. Aggregation Buffer: A New Parameter Block

Unlike the standard training strategy, where an augmentation function is used during training, we propose a two-step approach; given a GNN trained without DropEdge, we integrate $\mathrm{AGG}_B$ into each GNN layer and train $\mathrm{AGG}_B$ with DropEdge while freezing the pre-trained parameters. This two-step procedure provides several advantages:

1. **Practical Usability.** Our approach can be applied to any trained GNN. Separate training of $\mathrm{AGG}_B$ enables modular application even to already deployed models.

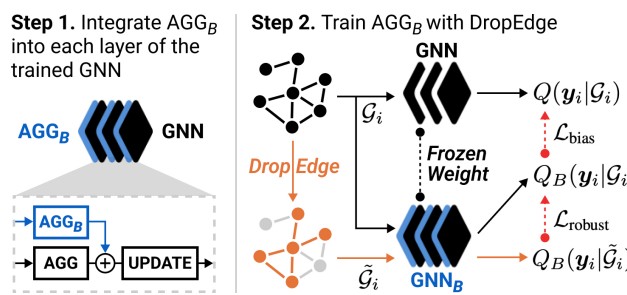

Figure 3. Illustration of $\mathrm{AGG}_B$ and its training scheme. After the integration into a pre-trained GNN, $\mathrm{AGG}_B$ is trained using $\mathcal{L}_\mathrm{RC}$ with DropEdge, while the pre-trained parameters remain frozen.

2. **Effectiveness.** Pre-training without DropEdge avoids the suboptimal minimization of the bias term observed in Section 3.3. As a result, $\mathrm{AGG}_B$ can focus entirely on optimizing the robustness term.

3. **No Knowledge Loss.** Freezing the pre-trained parameters prevents any unintended loss of knowledge during the training of $\mathrm{AGG}_B$. The integration of $\mathrm{AGG}_B$ can even be detached to get the original model back.

The main idea of our approach is to assign distinct roles to different sets of parameters: the pre-trained parameters focus on solving the primary classification task, while $\mathrm{AGG}_B$ is dedicated to mitigate representation changes caused by inconsistent graph structures. Given $\mathrm{AGG}_B$, while its details will be discussed later, we modify a GNN layer as

$$\boldsymbol{H}_\mathcal{N}^{(l)} = \mathrm{AGG}^{(l)}(\boldsymbol{H}^{(l-1)}, \boldsymbol{A}) + \mathrm{AGG}_B^{(l)}(\boldsymbol{H}^{(0:l-1)}, \boldsymbol{A}),$$

where $\mathrm{AGG}_B$ can leverage all available resources until the current layer $l$, including the adjacency matrix $\boldsymbol{A}$ and the preceding representations $\boldsymbol{H}^{(0:l-1)}$. We henceforth refer to the GNN model augmented with $\mathrm{AGG}_B$ as $\mathrm{GNN}_B$.

## 4.2. Essential Conditions for $\mathrm{AGG}_B$

The important part of our approach is to decide the actual function of $\mathrm{AGG}_B$. Existing methods for enhancing GNN layers, such as residual connections (He et al., 2016; Chen et al., 2020) and JKNet (Xu et al., 2018), are not considered as $\mathrm{AGG}_B$ since they fail to satisfy the essential conditions that $\mathrm{AGG}_B$ must meet to achieve its purpose. To derive our own approach that is better than existing methods, we first introduce the two essential conditions for $\mathrm{AGG}_B$.

---

**C1: Edge-Awareness.** When the adjacency matrix $\boldsymbol{A}$ is perturbed to $\tilde{\boldsymbol{A}}$, $\mathrm{AGG}_B$ should produce distinct outputs to compensate for structural changes:

$$\mathrm{AGG}_B^{(l)}(\boldsymbol{A}) \neq \mathrm{AGG}_B^{(l)}(\tilde{\boldsymbol{A}}).$$

---

This condition ensures that $\mathrm{AGG}_B$ adapts to structural variations by modifying its output accordingly. Existing layers

that depend only on node representations, such as residual connections and JKNet, fail to meet this condition as they produce identical outputs regardless of structural perturbations when the input representations remain the same.

---

**C2: Stability.** For any perturbed adjacency matrix $\tilde{A} \subset A$ created by random edge dropping, $\text{AGG}_B$ should produce outputs with a smaller deviation from the original output when given $A$, compared to when given $\tilde{A}$:

$$\|\text{AGG}_B^{(l)}(A)\|_{\text{F}} < \|\text{AGG}_B^{(l)}(\tilde{A})\|_{\text{F}}.$$

---

This condition ensures the knowledge learned by the original GNN to be preserved, contained in the frozen pre-trained parameters, by minimizing unnecessary changes under the original graph structure. At the same time, it provides sufficient flexibility to adapt and correct for structural perturbations, thereby optimizing edge-robustness without compromising the integrity of the original representations.

**Our Solution.** We propose a simple structure-aware form of $\text{AGG}_B$ which satisfies both conditions above:

$$g_B(H^{(0:l-1)}, A) = (D + I)^{-1} H^{(0:l-1)} W^{(l)},$$

where $D$ is the degree matrix of adjacency matrix $A$. Since it is *degree-normalized linear transformation*, its computation is faster than the regular AGG operation. When computed in parallel, integrating $\text{AGG}_B$ does not increase inference time, ensuring efficient execution.

**Theorem 4.1.** $g_B$ *satisfies the conditions C1 and C2.*

*Proof.* The proof is in Appendix D. □

### 4.3. Objective Function for Training $\text{AGG}_B$

We train the $\text{AGG}_B$ to minimize an objective function, $\mathcal{L}_{\text{RC}}$, referred to as the *robustness-controlled loss*, which has a few adjustments from $\tilde{\mathcal{L}}_{\text{Q}}$. First, we introduce a hyperparameter $\lambda$ to explicitly balance the strength between the bias term $\mathcal{L}_{\text{bias}}$ and the robustness term $\mathcal{L}_{\text{robust}}$:

$$\mathcal{L}_{\text{RC}}(\theta_B) = \mathcal{L}_{\text{bias}}(\theta_B) + \lambda \cdot \mathcal{L}_{\text{robust}}(\theta_B), \quad (5)$$

where $\theta_B$ refers to the set of parameters in $\text{AGG}_B$.

Then, we reformulate the bias term by replacing $P$ with $Q$. Since our method involves two-stage training, we can safely assume that the modeled distribution $Q$ of the pre-trained GNN is a good approximation of the true distribution $P$ at least within the training data. As a result, the bias term can simulate knowledge distillation in training data:

$$\mathcal{L}_{\text{bias}}(\theta_B) = \tfrac{1}{|V_{\text{trn}}|} \sum_{i \in V_{\text{trn}}} D_{\text{KL}}(Q(\boldsymbol{y}_i|\mathcal{G}_i)\|Q_B(\boldsymbol{y}_i|\mathcal{G}_i)),$$

$$\mathcal{L}_{\text{robust}}(\theta_B) = \tfrac{1}{|V|} \sum_{i \in V} D_{\text{KL}}(Q_B(\boldsymbol{y}_i|\mathcal{G}_i)\|Q_B(\boldsymbol{y}_i|\tilde{\mathcal{G}}_i)),$$

where $V_{\text{trn}}$ refers to the set of (labeled) training nodes, and $Q_B$ represents the modeled distribution of the GNN enhanced with $\text{AGG}_B$, which we refer to as $\text{GNN}_B$.

Unlike the bias term, the robustness term does not require access to the true distribution. This independence enables its application to all nodes, including unlabeled nodes, promoting comprehensive edge-robustness for the graph. On the other hand, it may not be effective to apply the bias term to all nodes as well, since it relies on an assumption that the pre-trained model distribution $Q$ approximates $P$ also in the unlabeled nodes, which is hardly true in practice.

## 5. Related Works

**Random Dropping Methods for GNNs.** Several random-dropping techniques were proposed for GNNs to improve their robustness, complementing the widely-used DropOut (Srivastava et al., 2014) method used in classical machine learning (You et al., 2020; Zhang et al., 2021; Li et al., 2023; Fang et al., 2023). DropEdge (Rong et al., 2020) removes a random subset of edges, while DropNode (Feng et al., 2020) removes nodes along with their connected edges. Existing graph sampling methods can also be seen as variants of these approaches. DropMessage (Fang et al., 2023) integrates DropNode, DropEdge, and DropOut by dropping propagated messages during the message-passing phase, offering higher information diversity. While these methods aim to reduce overfitting on edges in supervised learning, their performance improvements have been modest.

**Sub-optimalities of GNNs.** Incorporating edge information for its prediction is the core idea of GNNs. However, it also makes GNNs vulnerable to structural inconsistencies in the graph, making it suffer from well-known problems like degree bias and structural disparity. *Degree bias* refers to the tendency of performing significantly better on high-degree (head) nodes than on low-degree (tail) nodes (Tang et al., 2020; Liu et al., 2023). Tail-GNN (Liu et al., 2021) transfers representation-translation from head to tail nodes, while Coldbrew (Zheng et al., 2021) uses existing nodes as virtual neighbors for tail nodes. While both approaches improve the performance on tail nodes, they degrade the performance on head nodes and rely on manual degree thresholds. TUNEUP (Hu et al., 2023) fine-tunes GNNs with pseudo-labels and DropEdge, differing from our method by not freezing pre-trained parameters, lacking $\text{AGG}_B$, and using a different loss function. GraphPatcher (Ju et al., 2024) attaches virtual nodes to enhance the representations of tail nodes. *Structural disparity* arises when neighboring nodes have conflicting properties, such as heterophilous nodes in homophilous graphs. Recent studies (Wang et al., 2024; Zhu et al., 2020; Mao et al., 2024) show that MLPs outperform GNNs in such scenarios, implying that avoiding edge-reliance is often more beneficial. Our work addresses both issues holistically, improving GNN generalization by enhancing edge-robustness through the idea of $\text{AGG}_B$.

*Table 1.* Accuracy (%) of all models for test nodes grouped by degree. Head nodes refer to the top 33% of nodes by degree, while tail nodes refer to the bottom 33%. **Bold** values indicate the best performance, and underlined values indicate the second-best performance. Standard deviations are shown as subscripts. Our $GCN_B$ achieves at least the second-best in 31 out of 36 settings.

| Method | Cora | Citeseer | PubMed | Wiki-CS | A.Photo | A.Computer | CS | Physics | Arxiv | Actor | Squirrel | Chameleon |
|---|---|---|---|---|---|---|---|---|---|---|---|---|
| | | | | | | OVERALL PERFORMANCE | | | | | | |
| MLP | $64.86_{\pm1.21}$ | $65.55_{\pm0.76}$ | $84.62_{\pm0.28}$ | $75.98_{\pm0.51}$ | $85.97_{\pm0.81}$ | $80.81_{\pm0.40}$ | $\mathbf{93.55}_{\pm0.18}$ | $95.09_{\pm0.12}$ | $56.41_{\pm0.14}$ | $\mathbf{34.86}_{\pm0.97}$ | $32.55_{\pm1.51}$ | $32.10_{\pm3.10}$ |
| GCN | $83.44_{\pm1.44}$ | $72.45_{\pm0.80}$ | $86.48_{\pm0.17}$ | $80.26_{\pm0.34}$ | $92.21_{\pm1.36}$ | $88.24_{\pm0.63}$ | $91.85_{\pm0.29}$ | $95.18_{\pm0.17}$ | $71.80_{\pm0.10}$ | $30.16_{\pm0.73}$ | $41.67_{\pm2.42}$ | $40.19_{\pm4.29}$ |
| DropEdge | $83.27_{\pm1.55}$ | $72.29_{\pm0.60}$ | $86.47_{\pm0.21}$ | $80.22_{\pm0.55}$ | $92.14_{\pm1.42}$ | $88.08_{\pm1.08}$ | $91.91_{\pm0.16}$ | $95.13_{\pm0.16}$ | $71.73_{\pm0.21}$ | $29.86_{\pm0.82}$ | $38.40_{\pm2.57}$ | $\underline{40.51}_{\pm3.38}$ |
| DropNode | $83.65_{\pm1.83}$ | $72.20_{\pm0.67}$ | $86.55_{\pm0.18}$ | $80.11_{\pm0.61}$ | $91.89_{\pm1.21}$ | $88.17_{\pm0.40}$ | $91.93_{\pm0.28}$ | $95.11_{\pm0.16}$ | $71.72_{\pm0.16}$ | $29.07_{\pm0.93}$ | $38.01_{\pm2.00}$ | $39.74_{\pm2.79}$ |
| DropMessage | $83.45_{\pm1.56}$ | $72.44_{\pm0.76}$ | $\underline{86.56}_{\pm0.16}$ | $80.30_{\pm0.37}$ | $92.13_{\pm1.56}$ | $\underline{88.52}_{\pm0.44}$ | $92.08_{\pm0.21}$ | $95.14_{\pm0.18}$ | $71.93_{\pm0.20}$ | $29.62_{\pm1.05}$ | $38.75_{\pm3.34}$ | $40.48_{\pm3.07}$ |
| TUNEUP | $83.59_{\pm1.26}$ | $73.00_{\pm0.78}$ | $86.43_{\pm0.36}$ | $80.56_{\pm0.47}$ | $92.11_{\pm1.37}$ | $88.14_{\pm0.95}$ | $90.89_{\pm0.45}$ | $94.51_{\pm0.25}$ | $71.81_{\pm0.15}$ | $28.95_{\pm1.48}$ | $41.49_{\pm2.65}$ | $40.24_{\pm4.24}$ |
| GraphPatcher | $83.57_{\pm1.38}$ | $72.22_{\pm0.73}$ | $86.21_{\pm0.23}$ | $\underline{80.64}_{\pm0.51}$ | $\mathbf{92.89}_{\pm0.57}$ | $88.49_{\pm0.71}$ | $91.74_{\pm0.25}$ | $95.25_{\pm0.24}$ | $72.06_{\pm0.06}$ | $28.07_{\pm0.67}$ | $\underline{41.89}_{\pm2.49}$ | $40.35_{\pm4.11}$ |
| $GCN_B$(Ours) | $\mathbf{84.84}_{\pm1.39}$ | $\mathbf{73.32}_{\pm0.85}$ | $\mathbf{87.56}_{\pm0.27}$ | $\mathbf{80.75}_{\pm0.42}$ | $92.44_{\pm1.42}$ | $\mathbf{88.76}_{\pm0.65}$ | $\underline{93.54}_{\pm0.37}$ | $\mathbf{95.79}_{\pm0.17}$ | $\mathbf{72.43}_{\pm0.16}$ | $30.56_{\pm0.84}$ | $\mathbf{42.39}_{\pm2.19}$ | $\mathbf{40.96}_{\pm4.83}$ |
| | | | | | | ACCURACY ON HEAD NODES (HIGH-DEGREE) | | | | | | |
| MLP | $65.86_{\pm1.56}$ | $70.99_{\pm1.33}$ | $84.70_{\pm0.32}$ | $80.06_{\pm0.83}$ | $88.58_{\pm1.12}$ | $86.09_{\pm0.68}$ | $\mathbf{94.08}_{\pm0.24}$ | $97.50_{\pm0.14}$ | $63.93_{\pm0.17}$ | $\mathbf{34.27}_{\pm1.42}$ | $25.80_{\pm3.72}$ | $29.74_{\pm3.68}$ |
| GCN | $84.70_{\pm1.60}$ | $79.10_{\pm0.97}$ | $87.81_{\pm0.36}$ | $85.13_{\pm0.56}$ | $94.85_{\pm2.01}$ | $90.72_{\pm0.75}$ | $93.15_{\pm0.26}$ | $97.64_{\pm0.12}$ | $80.81_{\pm0.10}$ | $27.63_{\pm1.39}$ | $35.12_{\pm3.80}$ | $36.51_{\pm6.92}$ |
| DropEdge | $84.74_{\pm2.01}$ | $78.92_{\pm0.78}$ | $87.77_{\pm0.38}$ | $84.99_{\pm0.30}$ | $94.50_{\pm1.75}$ | $90.10_{\pm1.27}$ | $93.14_{\pm0.13}$ | $97.61_{\pm0.11}$ | $80.67_{\pm0.26}$ | $27.51_{\pm2.35}$ | $33.64_{\pm4.98}$ | $37.58_{\pm6.54}$ |
| DropNode | $84.82_{\pm2.47}$ | $79.01_{\pm1.34}$ | $87.80_{\pm0.54}$ | $85.02_{\pm0.55}$ | $94.53_{\pm2.02}$ | $90.53_{\pm0.54}$ | $93.19_{\pm0.23}$ | $97.59_{\pm0.11}$ | $80.73_{\pm0.25}$ | $26.62_{\pm1.15}$ | $32.33_{\pm5.09}$ | $35.92_{\pm6.81}$ |
| DropMessage | $84.86_{\pm1.60}$ | $79.33_{\pm1.10}$ | $87.84_{\pm0.45}$ | $84.96_{\pm0.42}$ | $94.62_{\pm2.24}$ | $91.01_{\pm0.75}$ | $93.28_{\pm0.29}$ | $97.57_{\pm0.11}$ | $80.77_{\pm0.25}$ | $27.55_{\pm1.70}$ | $30.42_{\pm4.14}$ | $\mathbf{38.85}_{\pm7.47}$ |
| TUNEUP | $84.58_{\pm1.46}$ | $\mathbf{79.43}_{\pm0.83}$ | $87.78_{\pm0.54}$ | $\underline{85.35}_{\pm0.51}$ | $94.73_{\pm1.95}$ | $90.62_{\pm1.12}$ | $92.12_{\pm0.40}$ | $97.26_{\pm0.15}$ | $80.74_{\pm0.18}$ | $26.56_{\pm1.43}$ | $34.85_{\pm3.81}$ | $35.82_{\pm5.38}$ |
| GraphPatcher | $\underline{85.21}_{\pm1.56}$ | $79.00_{\pm0.66}$ | $87.66_{\pm0.47}$ | $\mathbf{85.22}_{\pm0.65}$ | $\mathbf{95.28}_{\pm0.61}$ | $\mathbf{91.51}_{\pm0.69}$ | $93.25_{\pm0.42}$ | $97.46_{\pm0.20}$ | $\mathbf{80.89}_{\pm0.06}$ | $26.85_{\pm1.38}$ | $\mathbf{35.72}_{\pm4.41}$ | $36.40_{\pm4.99}$ |
| $GCN_B$(Ours) | $\mathbf{85.82}_{\pm1.31}$ | $\underline{79.41}_{\pm0.99}$ | $\mathbf{88.14}_{\pm0.60}$ | $85.04_{\pm0.56}$ | $94.84_{\pm2.05}$ | $90.70_{\pm0.80}$ | $\underline{93.87}_{\pm0.26}$ | $\mathbf{97.70}_{\pm0.11}$ | $80.85_{\pm0.13}$ | $27.65_{\pm1.48}$ | $35.38_{\pm4.28}$ | $37.68_{\pm7.39}$ |
| | | | | | | ACCURACY ON TAIL NODES (LOW-DEGREE) | | | | | | |
| MLP | $63.20_{\pm1.36}$ | $60.27_{\pm1.42}$ | $84.30_{\pm0.43}$ | $73.02_{\pm1.02}$ | $81.91_{\pm0.90}$ | $75.51_{\pm0.73}$ | $\underline{92.96}_{\pm0.28}$ | $92.76_{\pm0.21}$ | $49.71_{\pm0.19}$ | $\mathbf{34.47}_{\pm1.34}$ | $35.59_{\pm3.33}$ | $28.94_{\pm5.09}$ |
| GCN | $79.79_{\pm1.75}$ | $65.77_{\pm1.49}$ | $85.14_{\pm0.25}$ | $77.83_{\pm0.58}$ | $87.98_{\pm0.88}$ | $83.35_{\pm0.92}$ | $90.04_{\pm0.53}$ | $92.74_{\pm0.33}$ | $62.76_{\pm0.21}$ | $32.33_{\pm2.79}$ | $45.85_{\pm4.69}$ | $37.17_{\pm6.51}$ |
| DropEdge | $79.61_{\pm1.56}$ | $65.54_{\pm1.32}$ | $85.21_{\pm0.34}$ | $77.99_{\pm0.55}$ | $88.13_{\pm1.01}$ | $\underline{83.65}_{\pm1.13}$ | $90.09_{\pm0.32}$ | $92.66_{\pm0.36}$ | $62.65_{\pm0.33}$ | $31.94_{\pm1.91}$ | $43.20_{\pm3.17}$ | $34.91_{\pm5.93}$ |
| DropNode | $80.19_{\pm1.63}$ | $65.50_{\pm1.28}$ | $85.33_{\pm0.24}$ | $77.62_{\pm0.67}$ | $87.69_{\pm1.01}$ | $83.23_{\pm0.54}$ | $90.12_{\pm0.54}$ | $92.67_{\pm0.34}$ | $62.69_{\pm0.17}$ | $30.77_{\pm1.51}$ | $42.76_{\pm2.09}$ | $34.33_{\pm5.88}$ |
| DropMessage | $79.71_{\pm1.86}$ | $65.75_{\pm1.42}$ | $85.31_{\pm0.30}$ | $77.90_{\pm0.56}$ | $88.07_{\pm1.03}$ | $83.61_{\pm0.52}$ | $90.35_{\pm0.32}$ | $92.72_{\pm0.38}$ | $63.20_{\pm0.18}$ | $30.73_{\pm2.05}$ | $44.44_{\pm6.24}$ | $34.66_{\pm6.55}$ |
| TUNEUP | $80.40_{\pm1.77}$ | $66.35_{\pm1.66}$ | $85.12_{\pm0.28}$ | $78.13_{\pm0.80}$ | $87.87_{\pm0.97}$ | $83.45_{\pm0.86}$ | $88.98_{\pm0.59}$ | $91.64_{\pm0.32}$ | $62.89_{\pm0.19}$ | $31.09_{\pm3.29}$ | $45.51_{\pm4.66}$ | $\underline{37.50}_{\pm6.91}$ |
| GraphPatcher | $81.13_{\pm1.91}$ | $65.39_{\pm1.17}$ | $84.98_{\pm0.24}$ | $78.88_{\pm0.99}$ | $\mathbf{89.28}_{\pm0.66}$ | $83.24_{\pm1.02}$ | $89.48_{\pm0.49}$ | $93.03_{\pm0.39}$ | $63.56_{\pm0.13}$ | $29.22_{\pm1.71}$ | $46.24_{\pm3.85}$ | $\mathbf{38.29}_{\pm6.88}$ |
| $GCN_B$(Ours) | $\mathbf{82.05}_{\pm1.75}$ | $\mathbf{67.17}_{\pm1.37}$ | $\mathbf{86.85}_{\pm0.22}$ | $\mathbf{79.25}_{\pm0.58}$ | $\underline{88.53}_{\pm1.09}$ | $\mathbf{84.61}_{\pm0.98}$ | $\mathbf{92.97}_{\pm0.69}$ | $\mathbf{94.07}_{\pm0.27}$ | $\mathbf{64.40}_{\pm0.20}$ | $32.25_{\pm1.99}$ | $\mathbf{47.06}_{\pm4.13}$ | $37.35_{\pm6.99}$ |

# 6. Experiments

**Datasets.** We evaluate the accuracy of node classification for 12 widely-used benchmark graphs, including Cora, Citeseer, Pubmed, Computers, Photo, CS, Physics, Ogbn-arxiv, Actor, Squirrel and Chameleon (Shchur et al., 2018; Hu et al., 2020a; Pei et al., 2020). For Squirrel and Chameleon, we use the filtered versions following prior work (Platonov et al., 2023). These datasets are frequently used in prior works (Ju et al., 2024) and cover graphs from diverse domains with varying characteristics. Detailed descriptions of these datasets are provided in Appendix A.

**Baselines.** We compare $GCN_B$ (GCN with $AGG_B$) with its pre-trained model GCN (Kipf & Welling, 2017) as well as closely related graph learning methods, grouped into two categories. The first category includes random-dropping data augmentation algorithms for GNNs such as DropEdge (Rong et al., 2020), DropNode (Feng et al., 2020), and DropMessage (Fang et al., 2023). The second category includes state-of-the-art methods for addressing the degree bias problem such as TUNEUP (Hu et al., 2023) and GraphPatcher (Ju et al., 2024), which are relevant since degree-robustness can be seen as a special case of edge-robustness. Lastly, we include standard MLPs, providing a baseline for full edge-robustness as no edge-information is utilized.

**Setup.** We adopt the public dataset splits for Ogbn-arxiv (Hu et al., 2020a), Actor, Squirrel and Chameleon (Pei et al., 2020; Platonov et al., 2023). For the remaining eight datasets, we use an independently randomized 10%/10%/80% split for training, validation, and test, respectively. Our experiments are conducted using a two-layer GCN (Kipf & Welling, 2017), with hyperparameters selected via grid search based on validation accuracy across five runs, following prior works (Luo et al., 2024). For all baselines, we choose the hyperparameters as reported in the respective works or perform grid searches if no reference is available. Detailed hyperparameter settings and search spaces can be found in Appendix B.

**Evaluation.** All reported performances, including the ablation studies, are averaged over ten independent runs with different random seeds and splits, where we provide both the means and standard deviations. To assess edge-robustness, we evaluate the performance w.r.t. degree bias and structural disparity. For degree bias, we provide the performance on head nodes (the top 33% of nodes by degree) and tail nodes (bottom 33%), as defined in prior works (Ju et al., 2024). For structural disparity, nodes are grouped similarly based on their homophily ratio. Homophilous nodes are the top 33% with the highest homophily ratios, while heterophilous nodes comprise the bottom 33% with the lowest ratios.

## 6.1. Overall Performance

We compare $GCN_B$ with several baselines and present the results in Table 1. In terms of overall performance, $GCN_B$ achieves the highest accuracy in 9 and the second-best in 3

*Table 2.* Performance (%) of all models grouped by node homophily ratio. Homophilous nodes represent the top 33% of nodes with the highest homophily ratio, while heterophilous nodes represent the bottom 33%. **Bold** values indicate the best performance, and underlined values indicate the second-best performance. Standard deviations are shown as subscripts.

| Method | Cora | Citeseer | PubMed | Wiki-CS | Photo | Computer | CS | Physics | Arxiv | Actor | Squirrel | Chameleon |
|---|---|---|---|---|---|---|---|---|---|---|---|---|
| | | | | | ACCURACY ON HOMOPHILOUS NODES | | | | | | | |
| MLP | $71.68_{\pm1.77}$ | $76.37_{\pm1.19}$ | $89.90_{\pm0.58}$ | $86.30_{\pm0.58}$ | $83.63_{\pm1.41}$ | $86.01_{\pm0.59}$ | $96.96_{\pm0.25}$ | $98.02_{\pm0.07}$ | $74.69_{\pm0.14}$ | $36.88_{\pm1.52}$ | $35.84_{\pm2.73}$ | $33.50_{\pm5.96}$ |
| GCN | $92.69_{\pm1.53}$ | $87.96_{\pm1.25}$ | $95.99_{\pm0.22}$ | $94.05_{\pm0.65}$ | $96.45_{\pm3.76}$ | $94.47_{\pm0.53}$ | $99.25_{\pm0.15}$ | $99.32_{\pm0.15}$ | $95.43_{\pm0.08}$ | $\mathbf{39.47_{\pm1.62}}$ | $48.71_{\pm3.57}$ | $\underline{47.15_{\pm5.79}}$ |
| DropEdge | $92.38_{\pm1.88}$ | $88.06_{\pm0.90}$ | $96.12_{\pm0.36}$ | $94.44_{\pm0.42}$ | $96.35_{\pm3.84}$ | $\underline{94.72_{\pm0.43}}$ | $99.28_{\pm0.13}$ | $99.32_{\pm0.12}$ | $95.68_{\pm0.15}$ | $38.30_{\pm1.08}$ | $41.25_{\pm4.34}$ | $42.39_{\pm4.22}$ |
| DropNode | $92.81_{\pm1.63}$ | $87.84_{\pm0.97}$ | $96.17_{\pm0.32}$ | $93.99_{\pm0.60}$ | $96.48_{\pm3.71}$ | $94.29_{\pm0.30}$ | $\underline{99.31_{\pm0.17}}$ | $99.29_{\pm0.13}$ | $95.62_{\pm0.13}$ | $38.00_{\pm0.79}$ | $40.73_{\pm5.13}$ | $42.67_{\pm5.93}$ |
| DropMessage | $92.51_{\pm1.67}$ | $88.06_{\pm1.02}$ | $\mathbf{96.18_{\pm0.23}}$ | $94.49_{\pm0.34}$ | $96.35_{\pm3.67}$ | $94.62_{\pm0.47}$ | $\mathbf{99.37_{\pm0.15}}$ | $99.32_{\pm0.13}$ | $\mathbf{95.79_{\pm0.06}}$ | $38.02_{\pm1.64}$ | $45.22_{\pm4.60}$ | $41.25_{\pm4.98}$ |
| TUNEUP | $93.17_{\pm1.60}$ | $\underline{88.35_{\pm1.12}}$ | $96.04_{\pm0.30}$ | $94.05_{\pm0.65}$ | $96.30_{\pm3.76}$ | $94.57_{\pm0.55}$ | $99.14_{\pm0.14}$ | $99.26_{\pm0.13}$ | $95.67_{\pm0.11}$ | $37.94_{\pm2.57}$ | $48.58_{\pm3.79}$ | $\mathbf{47.89_{\pm5.89}}$ |
| GraphPatcher | $93.23_{\pm1.24}$ | $87.38_{\pm1.09}$ | $96.04_{\pm0.28}$ | $94.08_{\pm0.53}$ | $\mathbf{98.18_{\pm0.19}}$ | $94.65_{\pm0.64}$ | $98.33_{\pm0.30}$ | $\mathbf{99.44_{\pm0.07}}$ | $95.72_{\pm0.09}$ | $34.76_{\pm1.18}$ | $\underline{48.71_{\pm2.89}}$ | $44.83_{\pm5.22}$ |
| GCN$_B$(Ours) | $\mathbf{94.13_{\pm1.39}}$ | $\mathbf{88.78_{\pm1.52}}$ | $95.83_{\pm0.28}$ | $\mathbf{94.65_{\pm0.57}}$ | $\underline{96.54_{\pm3.76}}$ | $\mathbf{95.30_{\pm0.49}}$ | $98.64_{\pm0.56}$ | $\underline{99.33_{\pm0.17}}$ | $95.66_{\pm0.13}$ | $38.26_{\pm1.90}$ | $\mathbf{49.52_{\pm3.40}}$ | $47.01_{\pm5.60}$ |
| | | | | | ACCURACY ON HETEROPHILOUS NODES | | | | | | | |
| MLP | $50.93_{\pm0.98}$ | $\mathbf{44.88_{\pm1.82}}$ | $\mathbf{73.22_{\pm0.71}}$ | $57.90_{\pm0.93}$ | $81.71_{\pm1.34}$ | $66.84_{\pm0.69}$ | $\mathbf{85.96_{\pm0.29}}$ | $89.08_{\pm0.37}$ | $\mathbf{34.53_{\pm0.18}}$ | $\mathbf{31.66_{\pm2.79}}$ | $32.56_{\pm4.13}$ | $29.53_{\pm4.83}$ |
| GCN | $64.18_{\pm2.49}$ | $41.96_{\pm1.24}$ | $67.34_{\pm0.47}$ | $51.89_{\pm1.08}$ | $\underline{81.74_{\pm0.75}}$ | $71.42_{\pm1.25}$ | $76.81_{\pm0.68}$ | $86.60_{\pm0.37}$ | $32.51_{\pm0.28}$ | $19.13_{\pm1.55}$ | $42.19_{\pm5.54}$ | $33.74_{\pm7.61}$ |
| DropEdge | $64.09_{\pm2.68}$ | $41.78_{\pm1.27}$ | $67.12_{\pm0.52}$ | $50.97_{\pm1.49}$ | $81.50_{\pm0.69}$ | $71.06_{\pm1.95}$ | $76.92_{\pm0.35}$ | $86.47_{\pm0.40}$ | $31.70_{\pm0.52}$ | $19.29_{\pm1.72}$ | $41.59_{\pm6.04}$ | $37.01_{\pm5.02}$ |
| DropNode | $64.60_{\pm3.58}$ | $41.59_{\pm1.08}$ | $67.24_{\pm0.51}$ | $51.66_{\pm1.21}$ | $80.67_{\pm0.97}$ | $71.38_{\pm1.21}$ | $76.93_{\pm0.65}$ | $86.46_{\pm0.39}$ | $31.91_{\pm0.57}$ | $18.93_{\pm1.02}$ | $41.54_{\pm4.52}$ | $36.78_{\pm5.27}$ |
| DropMessage | $64.39_{\pm2.77}$ | $41.84_{\pm0.84}$ | $67.23_{\pm0.39}$ | $51.48_{\pm0.98}$ | $81.65_{\pm0.82}$ | $71.87_{\pm0.98}$ | $77.27_{\pm0.51}$ | $86.47_{\pm0.44}$ | $32.29_{\pm0.46}$ | $19.49_{\pm1.18}$ | $40.79_{\pm4.68}$ | $\mathbf{37.90_{\pm7.63}}$ |
| TUNEUP | $63.59_{\pm2.36}$ | $42.74_{\pm1.07}$ | $67.09_{\pm0.89}$ | $52.50_{\pm0.72}$ | $81.58_{\pm0.87}$ | $71.03_{\pm2.17}$ | $74.16_{\pm1.11}$ | $84.67_{\pm0.65}$ | $31.68_{\pm0.23}$ | $18.24_{\pm0.92}$ | $42.13_{\pm5.24}$ | $33.00_{\pm6.69}$ |
| GraphPatcher | $64.17_{\pm2.22}$ | $\underline{44.47_{\pm0.89}}$ | $66.41_{\pm0.34}$ | $53.03_{\pm0.86}$ | $81.46_{\pm1.68}$ | $71.56_{\pm1.96}$ | $78.67_{\pm0.53}$ | $86.87_{\pm0.64}$ | $33.38_{\pm0.14}$ | $18.49_{\pm1.33}$ | $\underline{42.41_{\pm5.16}}$ | $34.45_{\pm7.90}$ |
| GCN$_B$(Ours) | $\mathbf{65.54_{\pm2.34}}$ | $43.24_{\pm1.05}$ | $\underline{70.77_{\pm0.71}}$ | $52.44_{\pm1.27}$ | $\mathbf{82.29_{\pm1.05}}$ | $\mathbf{72.02_{\pm1.25}}$ | $82.75_{\pm0.63}$ | $88.43_{\pm0.35}$ | $\underline{34.02_{\pm0.34}}$ | $19.96_{\pm1.49}$ | $\mathbf{42.42_{\pm4.97}}$ | $35.03_{\pm7.53}$ |

out of 12 cases. Random-dropping methods such as DropEdge, DropNode, and DropMessage fail to consistently outperform GCN. This suggests that it is hard to enhance the edge-robustness of GNNs solely by performing data augmentation function, due to the inductive bias of GNNs as we have claimed in Section 3. Although GCN$_B$ is also based on DropEdge, it consistently improves the performance of GCN since it effectively addresses its edge-vulnerability.

One notable observation is that MLPs surpass all models in the CS and Actor datasets. This indicates that edges can contribute negatively to node classification depending on the structural property. On these datasets, our GCN$_B$ achieves the highest performance among all GNNs, demonstrating its effectiveness for enhancing edge-robustness.

TUNEUP and GraphPatcher generally improve the performance of GCN, demonstrating that addressing degree bias enhances the overall accuracy. However, their effectiveness compared to the base GCN is more limited than previously reported. Unlike previous works (Hu et al., 2023; Ju et al., 2024), which used basic hyperparameter settings, our experiments involve an extensive grid search to find optimal GCN configurations, making improvements more challenging. Despite this, GCN$_B$ significantly outperforms the base GCN, highlighting that edge-robustness is a critical factor for performance improvements in general.

### 6.2. Addressing Degree Bias

We assess the performance on head and tail nodes to evaluate the impact of our method on degree bias, as shown in Table 1. GCN$_B$ successfully mitigates degree bias, achieving at least the second-best accuracy on tail nodes in 10 and on head nodes in 9 datasets, demonstrating that enhancing edge-robustness effectively reduces degree bias. Random-

dropping approaches fail to consistently improve the tail performance. The models specifically designed for degree bias, TUNEUP and GraphPatcher, improve the tail performance but the improvement is relatively marginal.

Especially in heterophilous graphs (Actor and Squirrel), tail nodes generally outperform head nodes, contradicting typical degree bias trends. Degree-bias methods, which rely on the principle of transferring information from head to tail nodes, show limited effectiveness in these cases. However, GCN$_B$ still improves the performance by enhancing edge-robustness without being restricted to specific information flow, showing that edge-robustness is a broader focus.

### 6.3. Addressing Structural Disparity

We report the accuracy of the methods on homophilous and heterophilous nodes in Table 2 to evaluate structural disparity. Consistently with recent findings (Mao et al., 2024), our experiments show that MLPs generally outperform GNNs on heterophilous nodes, while GNNs perform better on homophilous nodes. Methods for addressing degree bias, particularly GraphPatcher, show some improvements on heterophilous nodes but inconsistently, indicating a correlation between degree bias and structural disparity, although the two problems seem distinct. GCN$_B$ achieves the highest GNN performance in heterophilous nodes on 9 datasets, demonstrating that enhancing edge-robustness can effectively mitigate structural disparity.

### 6.4. Generalization to Other GNN Architectures

An important advantage of AGG$_B$ is its broad applicability to most GNN architectures due to its modular design. To evaluate its versatility, we conduct extensive experiments

*Table 3.* Accuracy of different GNN models before and after the integration with $AGG_B$. $AGG_B$ achieves consistent and significant performance improvements across various architectures.

|  | Pubmed | CS | Arxiv | Chameleon |
|---|---|---|---|---|
| SAGE | $87.07_{\pm0.24}$ | $92.44_{\pm0.60}$ | $70.92_{\pm0.16}$ | $37.34_{\pm3.56}$ |
| $SAGE_B$ | $\mathbf{88.09_{\pm0.28}}$ | $\mathbf{93.36_{\pm0.47}}$ | $\mathbf{71.16_{\pm0.14}}$ | $\mathbf{37.85_{\pm3.80}}$ |
| GAT | $85.64_{\pm0.24}$ | $90.50_{\pm0.28}$ | $71.86_{\pm0.14}$ | $38.54_{\pm2.70}$ |
| $GAT_B$ | $\mathbf{87.47_{\pm0.37}}$ | $\mathbf{93.09_{\pm0.60}}$ | $\mathbf{72.26_{\pm0.14}}$ | $\mathbf{39.08_{\pm2.84}}$ |
| SGC | $84.01_{\pm0.76}$ | $90.89_{\pm0.45}$ | $69.15_{\pm0.05}$ | $38.24_{\pm3.00}$ |
| $SGC_B$ | $\mathbf{84.77_{\pm1.02}}$ | $\mathbf{91.90_{\pm0.43}}$ | $\mathbf{69.55_{\pm0.04}}$ | $\mathbf{38.91_{\pm3.08}}$ |
| GIN | $85.42_{\pm0.20}$ | $87.88_{\pm0.51}$ | $63.94_{\pm0.53}$ | $39.84_{\pm2.69}$ |
| $GIN_B$ | $\mathbf{87.18_{\pm0.17}}$ | $\mathbf{88.58_{\pm1.00}}$ | $\mathbf{65.66_{\pm0.75}}$ | $\mathbf{41.72_{\pm2.41}}$ |

*Table 4.* Accuracy with different layer architectures used as $AGG_B$, with none of the alternatives outperforming our proposed design.

|  | Pubmed | CS | Arxiv | Chameleon |
|---|---|---|---|---|
| JKNet | $87.45_{\pm0.25}$ | $93.36_{\pm0.56}$ | $72.19_{\pm0.18}$ | $40.29_{\pm4.68}$ |
| Residual | $87.46_{\pm0.24}$ | $92.05_{\pm0.28}$ | $72.29_{\pm0.12}$ | $39.77_{\pm4.57}$ |
| AGG | $86.82_{\pm0.55}$ | $91.63_{\pm0.28}$ | $72.27_{\pm0.12}$ | $40.69_{\pm2.69}$ |
| $AGG_B$ (ours) | $\mathbf{87.56_{\pm0.27}}$ | $\mathbf{93.54_{\pm0.37}}$ | $\mathbf{72.43_{\pm0.16}}$ | $\mathbf{40.96_{\pm4.83}}$ |

on four well-known architectures: SAGE (Hamilton et al., 2017), GAT (Veličković et al., 2018), SGC (Wu et al., 2019), and GIN (Xu et al., 2019). In Table 3, we compare the accuracy of each model before and after the integration of $AGG_B$. These results show that $AGG_B$ consistently delivers significant performance improvements across all architectures, demonstrating its wide applicability and effectiveness.

## 7. Ablation Studies

**Different Layer Architectures.** In line with Section 4.2, we evaluate alternative layer architectures for $AGG_B$, including residual connections, JKNet, and the AGG layer of GCN, while not changing other components of our method. Table 4 shows that our proposed design consistently outperforms these alternatives. Especially, the standard AGG shows no improvement on Pubmed and even degrades performance on CS from the base GCN. This suggests that using an additional AGG operation to resolve structural inconsistencies is ineffective as it introduces another inconsistency, as described in Theorem 3.9. These results highlight the importance of the conditions we propose in Section 4.2 for designing effective $AGG_B$. Comprehensive results across all datasets, accompanied by an in-depth discussion of this ablation study, are presented in Appendix G.

**Different Loss Functions.** Following Section 4.3, we explore alternative loss functions for training $AGG_B$. First, we evaluate a variant of $\mathcal{L}_{RC}$ by restricting the robustness term to the training set. While this approach is less effective than the proposed loss on all four datasets, it still consistently

*Table 5.* Accuracy with alternative loss functions to train $AGG_B$.

|  | Pubmed | CS | Arxiv | Chameleon |
|---|---|---|---|---|
| Pseudo-label | $86.62_{\pm0.37}$ | $92.48_{\pm0.15}$ | $71.84_{\pm0.18}$ | $40.14_{\pm4.01}$ |
| Self-distillation | $86.15_{\pm0.35}$ | $92.02_{\pm0.25}$ | $72.18_{\pm0.19}$ | $40.22_{\pm4.22}$ |
| Cross-entropy | $86.67_{\pm0.16}$ | $93.29_{\pm0.13}$ | $71.94_{\pm0.13}$ | $40.76_{\pm4.19}$ |
| $\mathcal{L}_{RC}$ (train only) | $86.89_{\pm0.33}$ | $92.67_{\pm0.57}$ | $72.26_{\pm0.15}$ | $40.31_{\pm3.98}$ |
| $\mathcal{L}_{RC}$ (ours) | $\mathbf{87.56_{\pm0.27}}$ | $\mathbf{93.54_{\pm0.37}}$ | $\mathbf{72.43_{\pm0.16}}$ | $\mathbf{40.96_{\pm4.83}}$ |

*Table 6.* Accuracy with various architectural hyperparameters, $AGG_B$ significantly enhances performance in all configurations.

|  | Pubmed | | Arxiv | |
|---|---|---|---|---|
|  | GCN | $GCN_B$ | GCN | $GCN_B$ |
| **NUMBER OF LAYERS** | | | | |
| 2 | $86.48_{\pm0.17}$ | $\mathbf{87.56_{\pm0.27}}$ | $71.80_{\pm0.10}$ | $\mathbf{72.43_{\pm0.16}}$ |
| 4 | $84.82_{\pm0.34}$ | $\mathbf{87.36_{\pm0.37}}$ | $71.53_{\pm0.20}$ | $\mathbf{72.42_{\pm0.20}}$ |
| 6 | $83.46_{\pm0.24}$ | $\mathbf{86.64_{\pm0.58}}$ | $70.77_{\pm0.27}$ | $\mathbf{71.79_{\pm0.21}}$ |
| 8 | $82.68_{\pm0.19}$ | $\mathbf{86.18_{\pm0.62}}$ | $70.17_{\pm0.45}$ | $\mathbf{71.36_{\pm0.38}}$ |
| **HIDDEN DIMENSION SIZE** | | | | |
| 64 | $86.54_{\pm0.26}$ | $\mathbf{87.18_{\pm0.32}}$ | $70.12_{\pm0.12}$ | $\mathbf{70.43_{\pm0.10}}$ |
| 128 | $86.56_{\pm0.25}$ | $\mathbf{87.36_{\pm0.25}}$ | $70.92_{\pm0.14}$ | $\mathbf{71.24_{\pm0.13}}$ |
| 256 | $86.54_{\pm0.17}$ | $\mathbf{87.54_{\pm0.23}}$ | $71.39_{\pm0.08}$ | $\mathbf{71.87_{\pm0.12}}$ |
| 512 | $86.48_{\pm0.17}$ | $\mathbf{87.56_{\pm0.27}}$ | $71.80_{\pm0.10}$ | $\mathbf{72.43_{\pm0.16}}$ |
| **ACTIVATION FUNCTION** | | | | |
| ReLU | $86.48_{\pm0.17}$ | $\mathbf{87.56_{\pm0.27}}$ | $71.80_{\pm0.10}$ | $\mathbf{72.43_{\pm0.16}}$ |
| ELU | $86.51_{\pm0.19}$ | $\mathbf{86.95_{\pm0.26}}$ | $71.50_{\pm0.22}$ | $\mathbf{71.97_{\pm0.20}}$ |
| Sigmoid | $85.66_{\pm0.16}$ | $\mathbf{86.62_{\pm0.42}}$ | $71.54_{\pm0.14}$ | $\mathbf{72.05_{\pm0.20}}$ |
| Tanh | $85.28_{\pm0.19}$ | $\mathbf{86.12_{\pm0.17}}$ | $71.72_{\pm0.15}$ | $\mathbf{72.25_{\pm0.14}}$ |

improves performance over the base GCN. Additionally, we test other loss functions, including the cross entropy using labels, knowledge distillation from the pre-trained model (Hinton, 2015; Zhang et al., 2022), and pseudo-labeling (Lee et al., 2013; Hu et al., 2023). Although these alternatives lead to performance improvements when combined with $AGG_B$, their effectiveness and consistency are limited compared to our objective function $\mathcal{L}_{RC}$.

**Architectural Hyperparameters.** For broader applicability, we expect $AGG_B$ to enhance robustness across diverse architectural hyperparameters not only in 2-layer GNNs. In Table 6, we present experimental results with varying numbers of layers, hidden dimension sizes, and activation functions in GCN. $AGG_B$ consistently improves performance in all configurations, even when the base model performs poorly due to overly deep layers or small hidden sizes. Notably in deep networks, where the performance typically degrades due to oversmoothing, $AGG_B$ significantly mitigates this decline, suggesting that the lack of edge-robustness is a potential reason of oversmoothing. These results demonstrate that $AGG_B$ is broadly applicable to GNNs regardless of their architectural hyperparameters. Further experimental results involving deeper architectures and additional datasets are presented in Appendix H.

*Table 7.* Effect of each key component in the proposed method.

|            | Pubmed | CS | Arxiv | Chameleon |
|------------|--------|-----|-------|-----------|
| $GCN_B$ | $\mathbf{87.56_{\pm 0.27}}$ | $\mathbf{93.54_{\pm 0.37}}$ | $72.43_{\pm 0.16}$ | $\mathbf{40.96_{\pm 4.83}}$ |
| (-) Freezing | $87.50_{\pm 0.31}$ | $93.50_{\pm 0.41}$ | $\mathbf{72.91_{\pm 0.14}}$ | $40.77_{\pm 3.86}$ |
| (-) $AGG_B$ | $86.82_{\pm 0.60}$ | $91.67_{\pm 0.41}$ | $72.21_{\pm 0.11}$ | $40.35_{\pm 3.89}$ |
| (-) Pre-train | $87.22_{\pm 0.14}$ | $92.78_{\pm 0.17}$ | $71.15_{\pm 0.12}$ | $39.39_{\pm 3.06}$ |

**Effect of Each Component.** We study the effect of each key component of our approach in Table 7. First, the accuracy usually degrades without freezing the parameters of the pre-trained GNN, indicating that freezing these parameters prevents unintended loss of the original knowledge. Next, we test the performance without $AGG_B$, which is equivalent to fine-tuning the pre-trained GNN using $\mathcal{L}_{RC}$. This leads to a significant performance degradation, even performing worse than the pre-trained GNN on the CS dataset. This supports our theoretical findings that the original GNN architecture is inherently limited in optimizing the robustness term in $\tilde{\mathcal{L}}_Q$. Finally, training $GCN_B$ in an end-to-end manner without pre-training also leads to performance degradation, demonstrating that the two-step training approach effectively optimizes both the bias and robustness terms.

## 8. Conclusion

In this work, we revisited DropEdge, identifying a critical limitation—DropEdge fails to fully optimize its robustness objective during training. Our theoretical analysis revealed that this limitation arises from the inherent properties of the AGG operation in GNNs, which struggles to maintain consistent representations under structural perturbations. To address this issue, we proposed *Aggregation Buffer* ($AGG_B$), a new parameter block designed to improve the AGG operations of GNNs. By refining the aggregation process, $AGG_B$ effectively optimizes the robustness objective, making the model significantly stronger to structural variations. Experiments on 12 node classification benchmarks and various GNN architectures demonstrated significant performance gains driven by $AGG_B$, especially for the problems related to structural inconsistencies, such as degree bias and structural disparity. Despite its effectiveness, our approach has limitations as a two-step approach; its performance relies on pre-trained knowledge, as $AGG_B$ focuses primarily on improving robustness. A potential direction for future work is to design a framework that enables $AGG_B$ to be trained end-to-end, allowing simultaneous optimization of both bias and robustness without dependency on pre-training.

## Acknowledgements

This work was supported by the National Research Foundation of Korea (NRF) grant funded by the Korea government (MSIT) (RS-2024-00341425 and RS-2024-00406985).

## Impact Statement

In this paper, we aim to advance the field of graph neural networks. Our work is believed to improve the reliability and effectiveness of GNNs on various applications. We do not foresee any direct negative societal impacts.

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

## A. Dataset Overview and Training Configuration for Base GCN

We selected the 12 datasets based on prior work (Ju et al., 2024). For Squirrel and Chameleon, we use the filtered versions provided by (Platonov et al., 2023) via their public repository: `https://github.com/yandex-research/heterophilous-graphs`. The remaining 10 datasets are sourced from the Deep Graph Library (DGL) (Wang et al., 2019). All graphs are treated as undirected. Detailed statistics are provided in Table 8.

*Table 8.* Statistics of datasets and hyperparameters used for training the base 2-layer GCN.

|  | Cora | Citeseer | PubMed | Wiki-CS | Photo | Computer | CS | Physics | Arxiv | Actor | Squirrel | Chameleon |
|---|---|---|---|---|---|---|---|---|---|---|---|---|
| # nodes | 2,708 | 3,327 | 19,717 | 11,701 | 7,650 | 13,752 | 18,333 | 34,493 | 169,343 | 7,600 | 2,334 | 890 |
| # edges | 10,556 | 9,228 | 88,651 | 431,726 | 238,162 | 491,722 | 163,788 | 495,924 | 1,166,243 | 33,391 | 93,996 | 18,598 |
| # features | 1,433 | 3,703 | 500 | 300 | 745 | 767 | 6,805 | 8,415 | 128 | 932 | 2,089 | 2,325 |
| # classes | 7 | 6 | 3 | 10 | 8 | 10 | 15 | 5 | 40 | 5 | 5 | 5 |
| Homophily Ratio | 0.8100 | 0.7355 | 0.8024 | 0.6543 | 0.8272 | 0.7772 | 0.8081 | 0.9314 | 0.6542 | 0.2167 | 0.2072 | 0.2361 |
| Hidden Dim | 512 | 512 | 512 | 512 | 512 | 512 | 256 | 64 | 512 | 64 | 256 | 256 |
| Learning Rate | 1e-2 | 1e-2 | 1e-2 | 1e-2 | 1e-2 | 1e-2 | 1e-3 | 1e-3 | 1e-2 | 1e-3 | 1e-2 | 1e-2 |
| Weight Decay | 5e-4 | 5e-4 | 5e-4 | 5e-4 | 5e-4 | 5e-4 | 5e-4 | 5e-5 | 5e-5 | 5e-4 | 5e-4 | 5e-4 |
| Dropout | 0.5 | 0.7 | 0.7 | 0.3 | 0.5 | 0.3 | 0.2 | 0.5 | 0.5 | 0.7 | 0.2 | 0.2 |
| AGG Scheme | Sym | Sym | Sym | RW | Sym | Sym | Sym | Sym | RW | Sym | Sym | Sym |

To train the base GCN, we conduct a grid search across five independent runs for each dataset, selecting the best hyperparameter configuration based on the highest validation accuracy, following the search space outlined in (Luo et al., 2024). The search space included hidden dimensions [64, 256, 512], dropout ratios [0.2, 0.3, 0.5, 0.7], weight decay values [0, 5e-4, 5e-5], and learning rates [1e-2, 1e-3, 5e-3]. We use the Adam optimizer (Kingma & Ba, 2015) for training with early stopping based on validation accuracy, using a patience of 100 epochs across all datasets.

We also consider two GCN aggregation schemes following prior work (Ju et al., 2024): (i) symmetric normalization, typically used in transductive settings, formulated as

$$\text{AGG}^{(l)}(\boldsymbol{H}^{(l-1)}, \boldsymbol{A}) = \boldsymbol{D}^{-\frac{1}{2}} \boldsymbol{A} \boldsymbol{D}^{-\frac{1}{2}} \boldsymbol{H}^{(l-1)} \boldsymbol{W}^{(l)},$$

and (ii) random-walk normalization, commonly used in inductive settings, given by

$$\text{AGG}^{(l)}(\boldsymbol{H}^{(l-1)}, \boldsymbol{A}) = \boldsymbol{D}^{-1} \boldsymbol{A} \boldsymbol{H}^{(l-1)} \boldsymbol{W}^{(l)}.$$

We select the aggregation scheme that achieves higher validation accuracy.

For the Squirrel and Chameleon datasets, we observe significant performance degradation when using standard GCN architectures. Therefore, guided by recommendations from (Platonov et al., 2023), which highlights data leakage issues and proposes filtered versions of these datasets, we incorporate residual connections and layer normalization into GCN. For reproducibility, detailed hyperparameter settings used for training the base GCN on each dataset are provided in Table 8.

## B. Experiment Configurations for Baselines and Proposed Method

All experiments are conducted on an NVIDIA RTX A6000 GPU with 48 GB of memory. We sincerely thank all the authors of baseline methods for providing open-source implementations, which greatly facilitated reproducibility and comparison.

**MLP.** For MLPs, we perform a grid search using the exact same hyperparameter search space as the base GCN. This extensive search, which is often overlooked for MLPs, leads to a unique observation: well-tuned MLPs can outperform GNNs on certain datasets, such as Actor and CS.

**Random Dropping Methods .** For DropEdge (Rong et al., 2020), DropNode (Feng et al., 2020), and DropMessage (Fang et al., 2023), we use the official repository of DropMessage: `https://github.com/zjunet/DropMessage`, which offers a unified framework for empirical comparison of the random dropping techniques. We conduct a grid search over drop ratios from 0.1 to 1.0 in increments of 0.1 for each method.

**GraphPatcher.** For GraphPatcher (Ju et al., 2024), we use the official repository: `https://github.com/jumxglhf/GraphPatcher`. We adopt the provided hyperparameter settings for overlapping datasets. For the remaining datasets, we perform a hyperparameter search over five independent runs, following the search space suggested in the original paper.

*Table 9.* Hyperparameters used to train $\text{AGG}_B$ integrated with a pre-trained 2-layer GCN

|  | Cora | Citeseer | PubMed | Wiki-CS | Photo | Computer | CS | Physics | Arxiv | Actor | Squirrel | Chameleon |
|---|---|---|---|---|---|---|---|---|---|---|---|---|
| $\lambda$ | 0.5 | 1.0 | 0.5 | 0.5 | 1.0 | 0.5 | 0.1 | 1.0 | 0.5 | 0.5 | 0.1 | 0.1 |
| DropOut | 0.7 | 0.7 | 0.0 | 0.5 | 0.0 | 0.2 | 0.7 | 0.7 | 0.0 | 0.2 | 0.2 | 0.0 |
| DropEdge | 0.5 | 0.2 | 0.2 | 0.2 | 0.5 | 0.5 | 0.2 | 0.2 | 0.5 | 0.5 | 0.7 | 0.7 |

**TUNEUP.** For TUNEUP (Hu et al., 2023), as the official implementation is not publicly available, we implemented the method ourselves. For the second training stage of TUNEUP, we conduct a grid search over DropEdge ratios from 0.1 to 1.0, and use the same search space for learning rate, dropout ratio, and weight decay as in the base GCN. Although TUNEUP was also manually implemented by the GraphPatcher authors, our extensively tuned implementation consistently yields higher performance across the most of datasets.

**Aggregation Buffer.** $\text{AGG}_B$ is trained after being integrated into a pre-trained GNN. For training $\text{AGG}_B$, we use the Adam optimizer with a fixed learning rate of 1e-2 and weight decay of 0.0 across all datasets. It is noteworthy that further performance gains may be achievable by tuning these hyperparameters for each dataset individually. Since the hidden dimension and number of layers are determined by the pre-trained model, they are not tunable hyperparameters for $\text{AGG}_B$. Training of $\text{AGG}_B$ is early stopped based on validation accuracy, with a patience of 100 epochs across all datasets.

$\text{AGG}_B$ requires tuning on three key hyperparameters: the dropout ratio, DropEdge ratio, and the coefficient $\lambda$, which balances the bias and robustness terms in $\mathcal{L}_{\text{RC}}$, as described in Equation (5). The search space used for these hyperparameters in our experiments is as follows:

- $\lambda$ values: [1, 0.5, 0.1],
- DropEdge ratio: [0.2, 0.5, 0.7, 1.0],
- Dropout ratio: [0, 0.2, 0.5, 0.7].

For hyperparameter tuning, we follow the same process used for training the base GCN, conducting a search across five independent runs and selecting the configuration with the highest validation accuracy. To ensure reproducibility, we provide the detailed hyperparameters for training $\text{AGG}_B$ across datasets in Table 9, and release our implementation as open-source at `https://github.com/dooho00/agg-buffer`.

# C. Proofs in Section 3

## C.1. Proof of Lemma 3.4

Activation functions play a pivotal role in Graph Neural Networks (GNNs) by introducing non-linearity, which enables the network to model complex relationships within graph-structured data. Ensuring that these activation functions are Lipschitz continuous is essential for guaranteeing that similarly aggregated representation can result a simliar output after applying the activation function. In this section, we formally derive the Lipschitz continuity of three widely used activation functions: Rectified Linear Unit (ReLU), Sigmoid, and Gaussian Error Linear Unit(GELU).

### C.1.1. DEFINITION OF LIPSCHITZ CONTINUITY

A function $f : \mathbb{R} \to \mathbb{R}$ is said to be **Lipschitz continuous** if there exists a constant $L \geq 0$ such that for all $x, y \in \mathbb{R}$,

$$|f(x) - f(y)| \leq L|x - y|.$$

### C.1.2. RECTIFIED LINEAR UNIT (RELU)

The Rectified Linear Unit (ReLU) activation function is defined as:

$$\text{ReLU}(x) = \max(0, x).$$

To prove that ReLU is 1-Lipschitz continuous, we need to show that:

$$|\text{ReLU}(x) - \text{ReLU}(y)| \leq |x - y| \quad \forall x, y \in \mathbb{R}.$$

**Case 1:** $x \geq 0$ and $y \geq 0$

In this case,
$$\text{ReLU}(x) = x \quad \text{and} \quad \text{ReLU}(y) = y.$$

Thus,
$$|\text{ReLU}(x) - \text{ReLU}(y)| = |x - y| \leq |x - y|.$$

**Case 2:** $x < 0$ and $y < 0$

Here,
$$\text{ReLU}(x) = 0 \quad \text{and} \quad \text{ReLU}(y) = 0.$$

Therefore,
$$|\text{ReLU}(x) - \text{ReLU}(y)| = |0 - 0| = 0 \leq |x - y|.$$

**Case 3:** $x \geq 0$ and $y < 0$ (without loss of generality)

In this scenario,
$$\text{ReLU}(x) = x \quad \text{and} \quad \text{ReLU}(y) = 0.$$

Thus,
$$|\text{ReLU}(x) - \text{ReLU}(y)| = |x - 0| = |x| \leq |x - y|.$$
This inequality holds because $x \geq 0$ and $y < 0$, implying $|x| \leq |x - y|$.

In all cases, $|\text{ReLU}(x) - \text{ReLU}(y)| \leq |x - y|$. Therefore, ReLU is **1-Lipschitz continuous**.

### C.1.3. SIGMOID FUNCTION

The Sigmoid activation function is defined as:
$$\sigma(x) = \frac{1}{1 + e^{-x}}.$$

The derivative of the Sigmoid function is:
$$\sigma'(x) = \sigma(x)(1 - \sigma(x)).$$
Using the fact that $\forall x \in \mathbb{R}$, $\sigma(x) > 0$, $1 - \sigma(x) > 0$, we apply AM-GM inequality:
$$\frac{\sigma(x) + (1 - \sigma(x))}{2} = \frac{1}{2} \geq \sqrt{\sigma(x)(1 - \sigma(x))}.$$

Squaring both sides,
$$\left(\frac{1}{2}\right)^2 = \frac{1}{4} \geq \sigma(x)(1 - \sigma(x)).$$

Thus,

$$0 \leq \sigma'(x) = \sigma(x)(1 - \sigma(x)) \leq \frac{1}{4}.$$

By the Mean Value Theorem, for any $x, y \in \mathbb{R}$, there exists some $c$ between $x$ and $y$ such that:

$$|\sigma(x) - \sigma(y)| = |\sigma'(c)||x - y|.$$

Using that $|\sigma'(c)| \leq \frac{1}{4}$, we have for all $x, y \in \mathbb{R}$

$$|\sigma(x) - \sigma(y)| \leq \frac{1}{4}|x - y|.$$

Therfore, Sigmoid is $\frac{1}{4}$**-Lipschitz continuous**.

C.1.4. GAUSSIAN ERROR LINEAR UNIT(GELU)

The GELU activation function is expressed as:

$$\text{GELU}(x) = x\Phi(x),$$

where $\Phi(x)$ is the cumulative distribution function (CDF) of the standard normal distribution:

$$\Phi(x) = \frac{1}{2}\left(1 + \text{erf}\left(\frac{x}{\sqrt{2}}\right)\right).$$

First, we compute the derivative of $\text{GELU}(x)$:

$$\frac{d}{dx}\text{GELU}(x) = \Phi(x) + x\phi(x),$$

where $\phi(x)$ is the probability density function (PDF) of the standard normal distribution:

$$\phi(x) = \frac{1}{\sqrt{2\pi}}e^{-x^2/2}.$$

In order to show the boundedness of the derivative, we examine the second derivative:

$$\frac{d^2}{dx^2}\text{GELU}(x) = 2\phi(x) - x^2\phi(x).$$

Setting the second derivative equal to zero to find critical points:

$$\frac{d^2}{dx^2}\text{GELU}(x) = 0 \implies \phi(x)(2 - x^2) = 0.$$

Since $\phi(x) > 0$ for all $x \in \mathbb{R}$, the extrema of $\frac{d}{dx}\text{GELU}(x)$ occurs at $x = \pm\sqrt{2}$.

Hence, it is enough to examine the value of $\frac{d}{dx}\text{GELU}(x) = \Phi(x) + x\phi(x)$ at $\pm\infty, \pm\sqrt{2}$ :

$$\lim_{x\to\infty}\frac{d}{dx}\text{GELU}(x) = \lim_{x\to\infty}\Phi(x) + \lim_{x\to\infty}x\phi(x) = 1 + 0 = 1$$

$$\lim_{x\to-\infty}\frac{d}{dx}\text{GELU}(x) = \lim_{x\to-\infty}\Phi(x) + \lim_{x\to-\infty}x\phi(x) = 0 + 0 = 0$$

$$\frac{d}{dx}\text{GELU}(\sqrt{2}) = \Phi(\sqrt{2}) + \sqrt{2}\cdot\phi(\sqrt{2}) = \frac{1}{2}(1 + \text{erf}(1)) + \sqrt{2}\frac{1}{\sqrt{2\pi}}e^{-1} \approx 1.129$$

$$\frac{d}{dx}\text{GELU}(-\sqrt{2}) = \Phi(-\sqrt{2}) - \sqrt{2}\cdot\phi(-\sqrt{2}) = \frac{1}{2}(1 + \text{erf}(-1)) - \sqrt{2}\frac{1}{\sqrt{2\pi}}e^{-1} \approx -0.129$$

using that

$$\lim_{x\to\infty}\Phi(x) = 1, \lim_{x\to-\infty}\Phi(x) = 0, \lim_{x\to\pm\infty}x\phi(x) = \lim_{x\to\pm\infty}\frac{1}{\sqrt{2\pi}}xe^{-x^2/2} = 0$$

Thus,

$$\left|\frac{d}{dx}\text{GELU}(x)\right| \leq 1.13, \quad \forall x \in \mathbb{R}.$$

Therefore, GELU is **1.13-Lipschitz continuous**.

## C.2. Proof of Lemma 3.5

The spectral norm of a matrix satisfies the following sub-multiplicative property:

$$\|\boldsymbol{AB}\|_2 \leq \|\boldsymbol{A}\|_2 \|\boldsymbol{B}\|_2$$

Using this property, we establish the discrepancy bound for 1-layer propagation in standard neural networks. For two intermediate representations $\boldsymbol{H}_1^{(l)}$ and $\boldsymbol{H}_2^{(l)}$, we have:

$$\begin{aligned}
\|\boldsymbol{H}_1^{(l)} - \boldsymbol{H}_2^{(l)}\|_2 &= \|\sigma(\boldsymbol{H}_1^{(l-1)}\boldsymbol{W}^{(l)} + \boldsymbol{b}^{(l)}) - \sigma(\boldsymbol{H}_2^{(l-1)}\boldsymbol{W}^{(l)} + \boldsymbol{b}^{(l)})\|_2 \\
&\leq L_\sigma \|(\boldsymbol{H}_1^{(l-1)}\boldsymbol{W}^{(l)} + \boldsymbol{b}^{(l)}) - (\boldsymbol{H}_2^{(l-1)}\boldsymbol{W}^{(l)} + \boldsymbol{b}^{(l)})\|_2 \\
&\leq L_\sigma \|\boldsymbol{H}_1^{(l-1)} - \boldsymbol{H}_2^{(l-1)}\|_2 \|\boldsymbol{W}^{(l)}\|_2,
\end{aligned}$$

where $L_\sigma$ is the Lipschitz constant for activation function $\sigma$.

## C.3. Proof of Theorem 3.6

The discrepancy in the final output representation can be bounded recursively as follows:

$$\begin{aligned}
\|\boldsymbol{H}_1^{(L)} - \boldsymbol{H}_2^{(L)}\|_2 &\leq L_\sigma \|\boldsymbol{W}^{(L)}\|_2 \|\boldsymbol{H}_1^{(L-1)} - \boldsymbol{H}_2^{(L-1)}\|_2 \\
&\leq L_\sigma^2 \|\boldsymbol{W}^{(L)}\|_2 \|\boldsymbol{W}^{(L-1)}\|_2 \|\boldsymbol{H}_1^{(L-2)} - \boldsymbol{H}_2^{(L-2)}\|_2 \\
&\leq \cdots \\
&\leq L_\sigma^{(L-l)} \left(\prod_{i=l+1}^{L} \|\boldsymbol{W}^{(i)}\|_2\right) \|\boldsymbol{H}_1^{(l)} - \boldsymbol{H}_2^{(l)}\|_2 \\
&= C \|\boldsymbol{H}_1^{(l)} - \boldsymbol{H}_2^{(l)}\|_2,
\end{aligned}$$

where $C = L_\sigma^{(L-l)} \prod_{i=l+1}^{L} \|\boldsymbol{W}^{(i)}\|_2$ represents the cascade constant.

## C.4. Proof of Lemma 3.7

In this proof, we aim to show how minimizing the discrepancy at each aggregation step effectively bounds the final representation discrepancy. To ensure consistent analysis, we assume that the representation matrix $\boldsymbol{H}_*$ is normalized, satisfying $\|\boldsymbol{H}_*\|_2 \leq |V|$. By quantifying propagation of discrepancy across linear transformations and various aggregation operations, we demonstrate that controlling intermediate discrepancies reduce the discrepancy in the final output.

### C.4.1. REGULAR AGGREGATION

For regular aggregation, the representation discrepancy satisfies:

$$\begin{aligned}
&\|\text{AGG}^{(l)}(\boldsymbol{H}_1^{(l-1)}, \boldsymbol{A}_1) - \text{AGG}^{(l)}(\boldsymbol{H}_2^{(l-1)}, \boldsymbol{A}_2)\|_2 \\
&= \|\boldsymbol{A}_1 \boldsymbol{H}_1^{(l-1)}\boldsymbol{W}^{(l)} - \boldsymbol{A}_2 \boldsymbol{H}_2^{(l-1)}\boldsymbol{W}^{(l)}\|_2 \\
&= \|(\boldsymbol{A}_1 \boldsymbol{H}_1^{(l-1)} - \boldsymbol{A}_1 \boldsymbol{H}_2^{(l-1)} + \boldsymbol{A}_1 \boldsymbol{H}_2^{(l-1)} - \boldsymbol{A}_2 \boldsymbol{H}_2^{(l-1)})\boldsymbol{W}^{(l)}\|_2 \\
&\leq (\|\boldsymbol{A}_1 \boldsymbol{H}_1^{(l-1)} - \boldsymbol{A}_1 \boldsymbol{H}_2^{(l-1)}\|_2 + \|\boldsymbol{A}_1 \boldsymbol{H}_2^{(l-1)} - \boldsymbol{A}_2 \boldsymbol{H}_2^{(l-1)}\|_2)\|\boldsymbol{W}^{(l)}\|_2 \\
&\leq (\|\boldsymbol{A}_1\|_2 \|\boldsymbol{H}_1^{(l-1)} - \boldsymbol{H}_2^{(l-1)}\|_2 + \|\boldsymbol{A}_1 - \boldsymbol{A}_2\|_2 \|\boldsymbol{H}_2^{(l-1)}\|_2)\|\boldsymbol{W}^{(l)}\|_2 \\
&\leq \|\boldsymbol{A}_1\|_2 \|\boldsymbol{W}^{(l)}\|_2 \|\boldsymbol{H}_1^{(l-1)} - \boldsymbol{H}_2^{(l-1)}\|_2 + |V| \|\boldsymbol{W}^{(l)}\|_2 \|\boldsymbol{A}_1 - \boldsymbol{A}_2\|_2
\end{aligned}$$

This shows that if $\boldsymbol{A}_1 = \boldsymbol{A}_2$, the representation discrepancy is linearly bounded by the input difference.

### C.4.2. ROW-NORMALIZED AGGREGATION

For row-normalized aggregation, the representation discrepancy satisfies:

$$
\begin{aligned}
&\|\text{AGG}^{(l)}(\boldsymbol{H}_1^{(l-1)}, \boldsymbol{A}_1) - \text{AGG}^{(l)}(\boldsymbol{H}_2^{(l-1)}, \boldsymbol{A}_2)\|_2 \\
&= \|\boldsymbol{D}_1^{-1}\boldsymbol{A}_1\boldsymbol{H}_1^{(l-1)}\boldsymbol{W}^{(l)} - \boldsymbol{D}_2^{-1}\boldsymbol{A}_2\boldsymbol{H}_2^{(l-1)}\boldsymbol{W}^{(l)}\|_2 \\
&= \|(\boldsymbol{D}_1^{-1}\boldsymbol{A}_1\boldsymbol{H}_1^{(l-1)} - \boldsymbol{D}_1^{-1}\boldsymbol{A}_1\boldsymbol{H}_2^{(l-1)} + \boldsymbol{D}_1^{-1}\boldsymbol{A}_1\boldsymbol{H}_2^{(l-1)} - \boldsymbol{D}_2^{-1}\boldsymbol{A}_2\boldsymbol{H}_2^{(l-1)})\boldsymbol{W}^{(l)}\|_2 \\
&\le (\|\boldsymbol{D}_1^{-1}\boldsymbol{A}_1\boldsymbol{H}_1^{(l-1)} - \boldsymbol{D}_1^{-1}\boldsymbol{A}_1\boldsymbol{H}_2^{(l-1)}\|_2 + \|\boldsymbol{D}_1^{-1}\boldsymbol{A}_1\boldsymbol{H}_2^{(l-1)} - \boldsymbol{D}_2^{-1}\boldsymbol{A}_2\boldsymbol{H}_2^{(l-1)}\|_2)\|\boldsymbol{W}^{(l)}\|_2 \\
&\le (\|\boldsymbol{D}_1^{-1}\boldsymbol{A}_1\|_2\|\boldsymbol{H}_1^{(l-1)} - \boldsymbol{H}_2^{(l-1)}\|_2 + \|\boldsymbol{D}_1^{-1}\boldsymbol{A}_1 - \boldsymbol{D}_2^{-1}\boldsymbol{A}_2\|_2\|\boldsymbol{H}_2^{(l-1)}\|_2)\|\boldsymbol{W}^{(l)}\|_2 \\
&\le \|\boldsymbol{W}^{(l)}\|_2\|\boldsymbol{H}_1^{(l-1)} - \boldsymbol{H}_2^{(l-1)}\|_2 + |V|\|\boldsymbol{W}^{(l)}\|_2\|\boldsymbol{D}_1^{-1}\boldsymbol{A}_1 - \boldsymbol{D}_2^{-1}\boldsymbol{A}_2\|_2
\end{aligned}
$$

noting that $\|\boldsymbol{D}_1^{-1}\boldsymbol{A}_1\|_2 = 1$ because row-normalized matrices have their largest eigenvalue equal to 1. This demonstrates that the discrepancy is linearly bounded if $\boldsymbol{A}_1 = \boldsymbol{A}_2$.

### C.4.3. SYMMETRIC-NORMALIZED AGGREGATION

For symmetric-normalized aggregation, the representation discrepancy satisfies:

$$
\begin{aligned}
&\|\text{AGG}^{(l)}(\boldsymbol{H}_1^{(l-1)}, \boldsymbol{A}_1) - \text{AGG}^{(l)}(\boldsymbol{H}_2^{(l-1)}, \boldsymbol{A}_2)\|_2 \\
&= \|\boldsymbol{D}_1^{-\frac{1}{2}}\boldsymbol{A}_1\boldsymbol{D}_1^{-\frac{1}{2}}\boldsymbol{H}_1^{(l-1)}\boldsymbol{W}^{(l)} - \boldsymbol{D}_2^{-\frac{1}{2}}\boldsymbol{A}_2\boldsymbol{D}_2^{-\frac{1}{2}}\boldsymbol{H}_2^{(l-1)}\boldsymbol{W}^{(l)}\|_2 \\
&= \|(\boldsymbol{D}_1^{-\frac{1}{2}}\boldsymbol{A}_1\boldsymbol{D}_1^{-\frac{1}{2}}\boldsymbol{H}_1^{(l-1)} - \boldsymbol{D}_1^{-\frac{1}{2}}\boldsymbol{A}_1\boldsymbol{D}_1^{-\frac{1}{2}}\boldsymbol{H}_2^{(l-1)} + \boldsymbol{D}_1^{-\frac{1}{2}}\boldsymbol{A}_1\boldsymbol{D}_1^{-\frac{1}{2}}\boldsymbol{H}_2^{(l-1)} - \boldsymbol{D}_2^{-\frac{1}{2}}\boldsymbol{A}_2\boldsymbol{D}_2^{-\frac{1}{2}}\boldsymbol{H}_2^{(l-1)})\boldsymbol{W}^{(l)}\|_2 \\
&\le (\|\boldsymbol{D}_1^{-\frac{1}{2}}\boldsymbol{A}_1\boldsymbol{D}_1^{-\frac{1}{2}}\boldsymbol{H}_1^{(l-1)} - \boldsymbol{D}_1^{-\frac{1}{2}}\boldsymbol{A}_1\boldsymbol{D}_1^{-\frac{1}{2}}\boldsymbol{H}_2^{(l-1)}\|_2 + \|\boldsymbol{D}_1^{-\frac{1}{2}}\boldsymbol{A}_1\boldsymbol{D}_1^{-\frac{1}{2}}\boldsymbol{H}_2^{(l-1)} - \boldsymbol{D}_2^{-\frac{1}{2}}\boldsymbol{A}_2\boldsymbol{D}_2^{-\frac{1}{2}}\boldsymbol{H}_2^{(l-1)}\|_2)\|\boldsymbol{W}^{(l)}\|_2 \\
&\le (\|\boldsymbol{D}_1^{-\frac{1}{2}}\boldsymbol{A}_1\boldsymbol{D}_1^{-\frac{1}{2}}\|_2\|\boldsymbol{H}_1^{(l-1)} - \boldsymbol{H}_2^{(l-1)}\|_2 + \|\boldsymbol{D}_1^{-\frac{1}{2}}\boldsymbol{A}_1\boldsymbol{D}_1^{-\frac{1}{2}} - \boldsymbol{D}_2^{-\frac{1}{2}}\boldsymbol{A}_2\boldsymbol{D}_2^{-\frac{1}{2}}\|_2\|\boldsymbol{H}_2^{(l-1)}\|_2)\|\boldsymbol{W}^{(l)}\|_2 \\
&\le \|\boldsymbol{W}^{(l)}\|_2\|\boldsymbol{H}_1^{(l-1)} - \boldsymbol{H}_2^{(l-1)}\|_2 + |V|\|\boldsymbol{W}^{(l)}\|_2\|\boldsymbol{D}_1^{-\frac{1}{2}}\boldsymbol{A}_1\boldsymbol{D}_1^{-\frac{1}{2}} - \boldsymbol{D}_2^{-\frac{1}{2}}\boldsymbol{A}_2\boldsymbol{D}_2^{-\frac{1}{2}}\|_2
\end{aligned}
$$

where $\|\boldsymbol{D}_1^{-\frac{1}{2}}\boldsymbol{A}_1\boldsymbol{D}_1^{-\frac{1}{2}}\|_2 = 1$ due to its normalization. This follows from the fact that $\boldsymbol{I} - \boldsymbol{D}_1^{-\frac{1}{2}}\boldsymbol{A}_1\boldsymbol{D}_1^{-\frac{1}{2}}$ is positive semi-definite, and there exists $\boldsymbol{x} = \boldsymbol{D}_1^{\frac{1}{2}}\boldsymbol{y}$ such that $\boldsymbol{x}^\top\boldsymbol{x} = \boldsymbol{x}^\top\boldsymbol{D}_1^{-\frac{1}{2}}\boldsymbol{A}_1\boldsymbol{D}_1^{-\frac{1}{2}}\boldsymbol{x}$ where $\boldsymbol{y}$ is eigenvector for $\boldsymbol{D}_1 - \boldsymbol{A}_1$ with corresponding eigenvalue 0. This implies that if $\boldsymbol{A}_1 = \boldsymbol{A}_2$, the representation discrepancy is linearly bounded by the input difference, similar to the case of regular aggregation.

### C.5. Proof of Theorem 3.8

Before we proceed with the proof, let us first establish the following lemma.

**Lemma C.1.** *Let $\boldsymbol{A}, \boldsymbol{B} \in \mathbb{R}_+^{m \times m}$, be two distinct matrices ($\boldsymbol{A} \ne \boldsymbol{B}$), and let $\sigma : \mathbb{R}^{m \times n} \to \mathbb{R}^{m \times n}$ be a non-constant, continuous, element-wise function. Then there exists some $\boldsymbol{Z} \in \mathbb{R}^{m \times n}$ such that*

$$
\sigma(\boldsymbol{A}\boldsymbol{Z}) \ne \sigma(\boldsymbol{B}\boldsymbol{Z})
$$

*Equivalently, no such $\sigma$ can satisfy $\sigma(\boldsymbol{A}\boldsymbol{Z}) = \sigma(\boldsymbol{B}\boldsymbol{Z})$ for all $\boldsymbol{Z}$ if $\boldsymbol{A} \ne \boldsymbol{B}$.*

*Proof.* Since $\boldsymbol{A} \ne \boldsymbol{B}$, there exists at least one index $(i, j)$ such that $a_{ij} \ne b_{ij}$. Denote $a_{ij} = a$ and $b_{ij} = b$ ($a > 0, b > 0$). We will construct a particular $\boldsymbol{Z}$ that reveals the difference under $\sigma$. Define $\boldsymbol{Z}$ so that its $j$-th row is a scalar variable $z \in \mathbb{R}$ and all other entries are zero. Then, each $(i, l)$-entry of $\boldsymbol{A}\boldsymbol{Z}$ is $az$, while the corresponding entry of $\boldsymbol{B}\boldsymbol{Z}$ is $bz$ for $l = 1, \cdots, n$. Because $\sigma$ is applied element-wise, $\sigma(\boldsymbol{A}\boldsymbol{Z}) = \sigma(\boldsymbol{B}\boldsymbol{Z})$ implies $\sigma(az) = \sigma(bz)$. We analyze two cases:

- **Case 1:** $a = 0$ or $b = 0$: Without loss of generality, let $b = 0$. Then $\sigma(az) = \sigma(0)$ must hold for all $z$. This forces $\sigma$ to be the constant, contradicting the given assumption.

- **Case 2:** $a \neq 0$ and $b \neq 0$: Without loss of generality, let $a > b$. Then, $\sigma(az) = \sigma(bz)$ for all $z$ implies

$$\sigma(z) = \sigma\left(\frac{b}{a}z\right) = \sigma\left(\left(\frac{b}{a}\right)^2 z\right) = \cdots = \sigma\left(\left(\frac{b}{a}\right)^n z\right)$$

for any $n > 1$. Since $b/a < 1$, by the continuity of $\sigma$, $\sigma(z) = \lim_{n \to \infty} \sigma\left(\left(\frac{b}{a}\right)^n z\right) = \sigma(0)$. Hence $\sigma$ would be constant on that range, again contradicting the non-constant assumption.

Thus, in either case, we find that the hypothesis $\sigma(\boldsymbol{AZ}) = \sigma(\boldsymbol{BZ})$ for all $\boldsymbol{Z}$ forces $\sigma$ to be constant, which is a contradiction. Therefore, there must exist some $\boldsymbol{Z}$ for which $\sigma(\boldsymbol{AZ}) \neq \sigma(\boldsymbol{BZ})$. $\qquad\square$

Now, we use the above lemma to show that, if $\hat{\boldsymbol{A}}_1 \neq \hat{\boldsymbol{A}}_2$, then no constant $C$ can bound the difference of GCN outputs in terms of the difference of inputs. Suppose, for contradiction, that there exists $C > 0$ such that for every pair $\boldsymbol{H}_1^{(l-1)}, \boldsymbol{H}_2^{(l-1)}$ exists, the following holds:

$$\|\boldsymbol{H}_1^{(l)} - \boldsymbol{H}_2^{(l)}\|_2 = \|\sigma(\hat{\boldsymbol{A}}_1\boldsymbol{H}_1^{(l-1)}\boldsymbol{W}^{(l)}) - \sigma(\hat{\boldsymbol{A}}_2\boldsymbol{H}_2^{(l-1)}\boldsymbol{W}^{(l)})\|_2 \leq C\|\boldsymbol{H}_1^{(l-1)} - \boldsymbol{H}_2^{(l-1)}\|_2$$

In particular, consider the case where $\boldsymbol{H}_1^{(l-1)} = \boldsymbol{H}_2^{(l-1)}$. Then $\|\boldsymbol{H}_1^{(l-1)} - \boldsymbol{H}_2^{(l-1)}\|_2 = 0$, so the right-hand side of above equation is zero. Thus, we must have $\sigma(\hat{\boldsymbol{A}}_1\boldsymbol{H}_1^{(l-1)}\boldsymbol{W}^{(l)}) = \sigma(\hat{\boldsymbol{A}}_2\boldsymbol{H}_2^{(l-1)}\boldsymbol{W}^{(l)})$. However, by Lemma C.1, there exists a suitable choice of $\boldsymbol{H}$ so that $\boldsymbol{HW}^{(l)}$ corresponds to $\boldsymbol{Z}$ of the lemma, leading to $\sigma(\hat{\boldsymbol{A}}_1\boldsymbol{HW}^{(l)}) \neq \sigma(\hat{\boldsymbol{A}}_2\boldsymbol{HW}^{(l)})$. If $\boldsymbol{H}_1^{(l-1)} = \boldsymbol{H}_2^{(l-1)} = \boldsymbol{H}$, we have

$$\|\sigma(\hat{\boldsymbol{A}}_1\boldsymbol{HW}^{(l)}) - \sigma(\hat{\boldsymbol{A}}_2\boldsymbol{HW}^{(l)})\|_2 > 0 = C\|\boldsymbol{H} - \boldsymbol{H}\|_2$$

This contradiction shows that no such input-independent constant $C$ can exist.

### C.6. Proof of Theorem 3.9

In GNNs with row-normalized or symmetric-normalized aggregation, the propagation of representation discrepancy is bounded as:

$$\|\boldsymbol{H}_1^{(l)} - \boldsymbol{H}_2^{(l)}\|_2 \leq L_\sigma\|\mathrm{AGG}^{(l)}(\boldsymbol{H}_1^{(l-1)}, \boldsymbol{A}_1) - \mathrm{AGG}^{(l)}(\boldsymbol{H}_2^{(l-1)}, \boldsymbol{A}_2)\|_2$$
$$\leq L_\sigma\|\boldsymbol{W}^{(l)}\|_2\|\boldsymbol{H}_1^{(l-1)} - \boldsymbol{H}_2^{(l-1)}\|_2 + L_\sigma|V|\|\boldsymbol{W}^{(l)}\|_2\|\hat{\boldsymbol{A}}_1 - \hat{\boldsymbol{A}}_2\|_2$$

where $\hat{\boldsymbol{A}}$ represents the normalized adjacency matrix and $L_\sigma$ be the Lipschitz constant of the activation function. This result shows that adjacency matrix perturbations introduce an additional error term, which grows through layers.

## D. Proofs in Section 4

### D.1. Proof of Theorem 4.1

Note that once condition C2 is satisfied, then C1 automatically holds. Thus, it suffices to show

$$\|\boldsymbol{D}_a^{-1}\boldsymbol{M}\|_F < \|\tilde{\boldsymbol{D}}_a^{-1}\boldsymbol{M}\|_F, \quad \text{where } \boldsymbol{D}_a = \boldsymbol{D} + \boldsymbol{I}, \ \boldsymbol{M} = \boldsymbol{H}^{(0:l-1)}\boldsymbol{W}^{(l)}.$$

Since $\tilde{\boldsymbol{A}} \subset \boldsymbol{A}$ by edge removal, each diagonal entry $\tilde{d}_i$ of $\tilde{\boldsymbol{D}}_a$ satisfies $0 < \tilde{d}_i \leq d_i$. Thus $\frac{1}{\tilde{d}_i} \geq \frac{1}{d_i}$ for all $i$.

Write $\boldsymbol{D}_a^{-1}\boldsymbol{M}$ and $\tilde{\boldsymbol{D}}_a^{-1}\boldsymbol{M}$ row by row. If $M_{i,\cdot}$ denotes the $i$-th row of $\boldsymbol{M}$, then

$$(\boldsymbol{D}_a^{-1}\boldsymbol{M})_{i,\cdot} = \frac{1}{d_i}M_{i,\cdot}, \quad (\tilde{\boldsymbol{D}}_a^{-1}\boldsymbol{M})_{i,\cdot} = \frac{1}{\tilde{d}_i}M_{i,\cdot}.$$

The Frobenius norm is

$$\|\boldsymbol{D}_a^{-1}\boldsymbol{M}\|_F^2 = \sum_{i=1}^n \frac{1}{d_i^2}\|M_{i,\cdot}\|_2^2, \quad \|\tilde{\boldsymbol{D}}_a^{-1}\boldsymbol{M}\|_F^2 = \sum_{i=1}^n \frac{1}{\tilde{d}_i^2}\|M_{i,\cdot}\|_2^2.$$

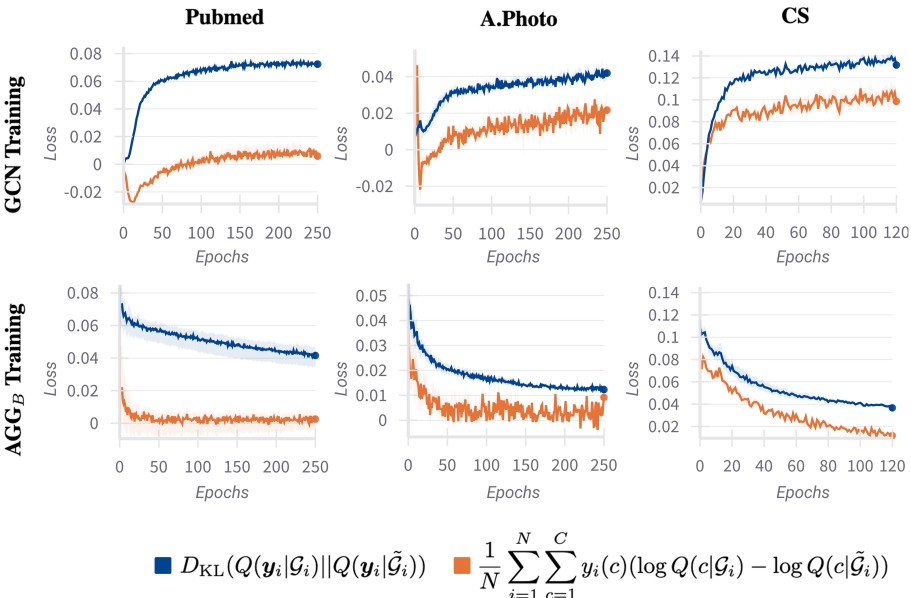

$$\blacksquare\ D_{\mathrm{KL}}(Q(\boldsymbol{y}_i|\mathcal{G}_i)\|Q(\boldsymbol{y}_i|\tilde{\mathcal{G}}_i)) \qquad \blacksquare\ \frac{1}{N}\sum_{i=1}^{N}\sum_{c=1}^{C} y_i(c)(\log Q(c|\mathcal{G}_i) - \log Q(c|\tilde{\mathcal{G}}_i))$$

*Figure 4.* Changes in two different approximations of the robustness loss, $\mathbb{E}_P[\log Q(\boldsymbol{y}_i|\mathcal{G}_i) - \log Q(\boldsymbol{y}_i|\tilde{\mathcal{G}}_i)]$, during training of the base GCN (top row) and AGG$_B$ (bottom row). Each curve represents the average over 10 independent runs, with shaded areas indicating the minimum and maximum values. Blue represents the robustness term in our proposed *robustness-controlled loss*, where $P$ is approximated by $Q$. Orange represents the label-based approximation, where P is approximated using ground-truth labels. Both approximations exhibit similar trends: robustness loss gradually emerges during GCN training and is further optimized during AGG$_B$ training.

Because $\frac{1}{\tilde{d}_i^2} \geq \frac{1}{d_i^2}$ whenever $\tilde{d}_i \leq d_i$, each term in the sum for $\tilde{\boldsymbol{D}}^{-1}\boldsymbol{M}$ is greater or equal. Therefore

$$\|\boldsymbol{D}_a^{-1}\boldsymbol{M}\|_F^2 \ \leq \ \|\tilde{\boldsymbol{D}}_a^{-1}\boldsymbol{M}\|_F^2 \quad \Longrightarrow \quad \|\boldsymbol{D}_a^{-1}\boldsymbol{M}\|_F \ \leq \ \|\tilde{\boldsymbol{D}}_a^{-1}\boldsymbol{M}\|_F.$$

Note that the equality holds if and only if $\|M_{j,\cdot}\|_2^2 = 0$ for every $j$'s such that $\tilde{d}_j < d_j$. Such a configuration occupies a measure-zero subset of whole space, and thus arises with probability zero in typical real-world scenarios.

Substituting back completes the proof:

$$\|\mathrm{AGG}_B^{(l)}(\boldsymbol{A})\|_{\mathrm{F}} = \|\boldsymbol{D}_a^{-1}\boldsymbol{M}\|_F \ < \ \|\tilde{\boldsymbol{D}}_a^{-1}\boldsymbol{M}\|_F = \|\mathrm{AGG}_B^{(l)}(\tilde{\boldsymbol{A}})\|_{\mathrm{F}}.$$

## E. Validity of the Approximation on Robustness Loss

Between equation 3 and equation 4, we approximate the robustness term in the shifted objective under DropEdge. Specifically, the expectation with respect to the true distribution $P$ is approximated using the model's predictive distribution $Q$ as follows:

$$\mathbb{E}_P[\log Q(\boldsymbol{y}_i|\mathcal{G}_i) - \log Q(\boldsymbol{y}_i|\tilde{\mathcal{G}}_i)] \approx D_{\mathrm{KL}}(Q(\boldsymbol{y}_i|\mathcal{G}_i)\|Q(\boldsymbol{y}_i|\tilde{\mathcal{G}}_i)).$$

This approximation is based on the assumption $Q \approx P$. Since the true distribution $P$ is inaccessible during training, this assumption allows the term to be computed in practice.

Although the assumption $Q \approx P$ may not strictly hold—particularly in the early stages of training—it becomes increasingly valid as training progresses. Since the model is trained using cross-entropy loss, it explicitly minimizes the KL divergence $D_{\mathrm{KL}}(P(y_i|\mathcal{G}_i)\|Q(y_i|\mathcal{G}_i))$, gradually aligning $Q$ with $P$ on the training distribution. Moreover, our framework employs a two-step training procedure, in which this approximation is utilized only after the base GCN has been trained. This staged design ensures that the approximation is applied under more reliable conditions, promoting stable and effective optimization of the proposed *robustness-controlled loss*, $\mathcal{L}_{\mathrm{RC}}$.

*Table 10.* Test accuracy under varying levels of random edge removal (%) across 12 datasets. A value of 100% indicates that no edges are removed, whereas 0% indicates complete edge removal. Bold entries denote the highest performance for each setting. $\text{AGG}_B$ significantly enhances robustness, outperforming the baselines (GCN, DropEdge) in 56 out of 60 cases.

| | Cora | | | Citeseer | | | Pubmed | | | Wiki-CS | | |
|---|---|---|---|---|---|---|---|---|---|---|---|---|
| | GCN | DropEdge | $\text{GCN}_B$ | GCN | DropEdge | $\text{GCN}_B$ | GCN | DropEdge | $\text{GCN}_B$ | GCN | DropEdge | $\text{GCN}_B$ |
| 100% | $83.44_{\pm1.44}$ | $83.27_{\pm1.55}$ | $\mathbf{84.84_{\pm1.39}}$ | $72.45_{\pm0.80}$ | $72.29_{\pm0.60}$ | $\mathbf{73.32_{\pm0.85}}$ | $86.48_{\pm0.17}$ | $86.47_{\pm0.21}$ | $\mathbf{87.56_{\pm0.27}}$ | $80.26_{\pm0.34}$ | $80.22_{\pm0.55}$ | $\mathbf{80.75_{\pm0.42}}$ |
| 75% | $81.85_{\pm1.28}$ | $81.78_{\pm1.53}$ | $\mathbf{84.53_{\pm1.38}}$ | $71.66_{\pm0.69}$ | $71.54_{\pm0.47}$ | $\mathbf{73.16_{\pm1.01}}$ | $85.99_{\pm0.20}$ | $86.01_{\pm0.12}$ | $\mathbf{87.52_{\pm0.33}}$ | $79.28_{\pm0.41}$ | $79.34_{\pm0.55}$ | $\mathbf{80.17_{\pm0.47}}$ |
| 50% | $79.63_{\pm1.65}$ | $79.09_{\pm1.51}$ | $\mathbf{84.31_{\pm1.37}}$ | $70.76_{\pm0.64}$ | $70.33_{\pm0.40}$ | $\mathbf{72.90_{\pm1.23}}$ | $85.48_{\pm0.22}$ | $85.69_{\pm0.22}$ | $\mathbf{87.51_{\pm0.26}}$ | $77.85_{\pm0.22}$ | $78.09_{\pm0.47}$ | $\mathbf{79.20_{\pm0.38}}$ |
| 25% | $76.28_{\pm1.67}$ | $76.48_{\pm1.06}$ | $\mathbf{84.44_{\pm1.59}}$ | $69.35_{\pm0.69}$ | $68.94_{\pm0.46}$ | $\mathbf{72.66_{\pm1.39}}$ | $84.80_{\pm0.22}$ | $84.90_{\pm0.24}$ | $\mathbf{87.43_{\pm0.26}}$ | $75.50_{\pm0.45}$ | $76.51_{\pm0.45}$ | $\mathbf{77.83_{\pm0.72}}$ |
| 0% | $72.63_{\pm1.82}$ | $72.33_{\pm1.70}$ | $\mathbf{84.17_{\pm1.50}}$ | $68.12_{\pm0.54}$ | $67.54_{\pm0.49}$ | $\mathbf{72.23_{\pm1.71}}$ | $83.86_{\pm0.38}$ | $84.18_{\pm0.39}$ | $\mathbf{86.81_{\pm0.31}}$ | $69.60_{\pm0.96}$ | $72.54_{\pm0.82}$ | $\mathbf{72.86_{\pm1.50}}$ |

| | A.Photo | | | A.Computer | | | CS | | | Physics | | |
|---|---|---|---|---|---|---|---|---|---|---|---|---|
| | GCN | DropEdge | $\text{GCN}_B$ | GCN | DropEdge | $\text{GCN}_B$ | GCN | DropEdge | $\text{GCN}_B$ | GCN | DropEdge | $\text{GCN}_B$ |
| 100% | $92.21_{\pm1.36}$ | $92.14_{\pm1.42}$ | $\mathbf{92.44_{\pm1.42}}$ | $88.24_{\pm0.63}$ | $88.08_{\pm1.08}$ | $\mathbf{88.76_{\pm0.65}}$ | $91.85_{\pm0.29}$ | $91.91_{\pm0.16}$ | $\mathbf{93.54_{\pm0.37}}$ | $95.18_{\pm0.17}$ | $95.13_{\pm0.16}$ | $\mathbf{95.79_{\pm0.17}}$ |
| 75% | $92.08_{\pm1.31}$ | $92.02_{\pm1.42}$ | $\mathbf{92.38_{\pm1.35}}$ | $88.01_{\pm0.59}$ | $87.83_{\pm0.99}$ | $\mathbf{88.71_{\pm0.66}}$ | $91.41_{\pm0.28}$ | $91.43_{\pm0.22}$ | $\mathbf{93.59_{\pm0.38}}$ | $94.88_{\pm0.19}$ | $94.87_{\pm0.16}$ | $\mathbf{95.77_{\pm0.16}}$ |
| 50% | $91.67_{\pm1.41}$ | $91.77_{\pm1.39}$ | $\mathbf{92.29_{\pm1.40}}$ | $87.41_{\pm0.57}$ | $87.38_{\pm1.04}$ | $\mathbf{88.54_{\pm0.55}}$ | $90.77_{\pm0.25}$ | $90.71_{\pm0.22}$ | $\mathbf{93.56_{\pm0.42}}$ | $94.53_{\pm0.17}$ | $94.54_{\pm0.20}$ | $\mathbf{95.76_{\pm0.17}}$ |
| 25% | $90.79_{\pm1.72}$ | $90.92_{\pm1.51}$ | $\mathbf{91.90_{\pm1.51}}$ | $86.20_{\pm0.54}$ | $86.28_{\pm0.88}$ | $\mathbf{88.02_{\pm0.55}}$ | $89.91_{\pm0.18}$ | $89.95_{\pm0.21}$ | $\mathbf{93.53_{\pm0.53}}$ | $93.99_{\pm0.18}$ | $93.96_{\pm0.18}$ | $\mathbf{95.72_{\pm0.17}}$ |
| 0% | $84.88_{\pm1.81}$ | $85.99_{\pm1.66}$ | $\mathbf{86.11_{\pm3.35}}$ | $76.77_{\pm1.48}$ | $78.17_{\pm1.32}$ | $\mathbf{82.50_{\pm1.19}}$ | $93.10_{\pm0.31}$ | $93.17_{\pm0.23}$ | $\mathbf{93.18_{\pm0.74}}$ | $94.33_{\pm0.51}$ | $94.63_{\pm0.43}$ | $\mathbf{95.44_{\pm0.20}}$ |

| | Arxiv | | | Actor | | | Squirrel | | | Chameleon | | |
|---|---|---|---|---|---|---|---|---|---|---|---|---|
| | GCN | DropEdge | $\text{GCN}_B$ | GCN | DropEdge | $\text{GCN}_B$ | GCN | DropEdge | $\text{GCN}_B$ | GCN | DropEdge | $\text{GCN}_B$ |
| 100% | $71.80_{\pm0.10}$ | $71.73_{\pm0.21}$ | $\mathbf{72.43_{\pm0.16}}$ | $30.16_{\pm0.73}$ | $29.86_{\pm0.82}$ | $\mathbf{30.56_{\pm0.84}}$ | $41.67_{\pm2.42}$ | $41.66_{\pm2.11}$ | $\mathbf{42.39_{\pm2.19}}$ | $40.19_{\pm4.29}$ | $40.23_{\pm4.34}$ | $\mathbf{40.96_{\pm4.83}}$ |
| 75% | $70.41_{\pm0.17}$ | $70.42_{\pm0.22}$ | $\mathbf{71.57_{\pm0.17}}$ | $\mathbf{30.72_{\pm0.98}}$ | $30.33_{\pm0.95}$ | $30.70_{\pm0.92}$ | $40.64_{\pm2.89}$ | $40.71_{\pm2.71}$ | $\mathbf{41.45_{\pm2.54}}$ | $39.52_{\pm4.37}$ | $39.51_{\pm4.21}$ | $\mathbf{40.16_{\pm4.86}}$ |
| 50% | $67.97_{\pm0.18}$ | $68.13_{\pm0.30}$ | $\mathbf{69.95_{\pm0.14}}$ | $31.09_{\pm1.21}$ | $30.49_{\pm0.99}$ | $\mathbf{31.26_{\pm1.08}}$ | $39.88_{\pm2.19}$ | $39.63_{\pm2.36}$ | $\mathbf{40.86_{\pm2.21}}$ | $39.93_{\pm4.22}$ | $\mathbf{40.13_{\pm5.18}}$ | $39.83_{\pm4.08}$ |
| 25% | $62.81_{\pm0.32}$ | $63.11_{\pm0.42}$ | $\mathbf{66.28_{\pm0.14}}$ | $\mathbf{31.24_{\pm0.91}}$ | $31.20_{\pm1.42}$ | $31.00_{\pm1.61}$ | $37.47_{\pm2.52}$ | $37.33_{\pm2.30}$ | $\mathbf{40.06_{\pm2.66}}$ | $38.41_{\pm3.20}$ | $38.46_{\pm3.68}$ | $\mathbf{38.57_{\pm4.39}}$ |
| 0% | $44.09_{\pm0.64}$ | $44.55_{\pm0.95}$ | $\mathbf{53.57_{\pm0.35}}$ | $\mathbf{33.68_{\pm1.38}}$ | $32.80_{\pm1.66}$ | $30.76_{\pm1.76}$ | $32.18_{\pm2.51}$ | $32.40_{\pm1.67}$ | $\mathbf{35.40_{\pm2.43}}$ | $33.24_{\pm3.13}$ | $33.22_{\pm3.10}$ | $\mathbf{36.29_{\pm4.64}}$ |

To empirically evaluate the validity of this approximation, we estimate the expectation under $P$ using ground-truth labels as a proxy. Specifically, we computed the following quantity:

$$\frac{1}{N}\sum_{i=1}^{N}\sum_{c=1}^{C} y_i(c)(\log Q(c|\mathcal{G}_i) - \log Q(c|\tilde{\mathcal{G}}_i)),$$

where $y_i(c)$ denotes the one-hot encoded ground-truth label for node $i$. While this proxy samples only one class per node and may be affected by label noise, it still offers a practical estimate for validating the approximation. As shown in Figure 4, the trend of this label-based quantity closely mirrors that of the approximated KL divergence, indicating that our approximation effectively captures the underlying behavior. Furthermore, $\text{AGG}_B$ exhibits robust optimization behavior even under this label-based approximation, demonstrating its effectiveness in terms of robustness optimization.

## F. Assessing Edge-Robustness via Random Edge Removal at Test Time

While we previously demonstrated the edge-robustness benefits of $\text{AGG}_B$ through improvements in degree bias and structural disparity, we now provide a more direct evaluation by measuring model performance under random edge removal during inference. Specifically, we assess how test accuracy degrades as edges are randomly removed from the input graph. We compare three models: (1) a standard GCN trained normally, (2) a GCN trained with DropEdge, and (3) $\text{GCN}_B$, which incorporates $\text{AGG}_B$ into a pre-trained GCN. The results are presented in Table 10.

$\text{GCN}_B$ significantly outperforms both DropEdge and standard GCN in 56 out of 60 cases, indicating that $\text{AGG}_B$ enables GCNs to generate more consistent representations under structural perturbations, thereby exhibiting superior edge-robustness.

Interestingly, in 3 of the 4 cases where $\text{GCN}_B$ does not outperform the baselines, the performance of the standard GCN improves as edges are removed—specifically on the Actor dataset. This aligns with our observation in Table 1 that an MLP outperforms GCN on this dataset, suggesting that leveraging edge information may not be beneficial. These findings imply that the edges in Actor are likely too noisy or uninformative. Nevertheless, even on Actor, $\text{GCN}_B$ maintains stable accuracy under edge removal, highlighting that $\text{AGG}_B$ still contributes to enhanced edge-robustness.

In contrast, models trained with DropEdge often show marginal improvements or even performance degradation compared to standard GCNs. This supports our claim that DropEdge alone is insufficient for achieving edge-robustness, due to the inherent inductive bias of GNNs.

*Table 11.* Accuracy across different layer architectures used as $\text{AGG}_B$. Blue indicates no improvement over the base GCN, and bold text highlights the best-performing architecture per dataset. The rightmost column reports the average rank. Our proposed design consistently improves performance and achieves the best overall ranking.

| Method | Cora | Citeseer | PubMed | Wiki-CS | A.Photo | A.Computer | CS | Physics | Arxiv | Actor | Squirrel | Chameleon | Rank |
|---|---|---|---|---|---|---|---|---|---|---|---|---|---|
| GCN | $83.44_{\pm 1.44}$ | $72.45_{\pm 0.80}$ | $86.48_{\pm 0.17}$ | $80.26_{\pm 0.34}$ | $92.21_{\pm 1.36}$ | $88.24_{\pm 0.63}$ | $91.85_{\pm 0.29}$ | $95.18_{\pm 0.17}$ | $71.80_{\pm 0.10}$ | $30.16_{\pm 0.73}$ | $41.67_{\pm 2.42}$ | $40.19_{\pm 4.29}$ | 4.75 |
| AGG | $84.01_{\pm 1.58}$ | $72.70_{\pm 0.82}$ | $86.82_{\pm 0.55}$ | $79.54_{\pm 0.94}$ | $91.74_{\pm 1.76}$ | $88.03_{\pm 0.74}$ | $91.63_{\pm 0.28}$ | $95.02_{\pm 0.31}$ | $72.27_{\pm 0.12}$ | $29.29_{\pm 1.01}$ | $40.18_{\pm 4.48}$ | $40.69_{\pm 2.69}$ | 4.92 |
| Residual | $83.29_{\pm 2.50}$ | $72.71_{\pm 0.83}$ | $87.46_{\pm 0.24}$ | $\mathbf{80.80_{\pm 1.08}}$ | $92.40_{\pm 1.64}$ | $\mathbf{88.95_{\pm 0.44}}$ | $92.05_{\pm 0.28}$ | $95.27_{\pm 0.38}$ | $72.29_{\pm 0.12}$ | $30.57_{\pm 0.53}$ | $41.32_{\pm 2.95}$ | $39.77_{\pm 4.57}$ | 3.00 |
| JKNet-style | $83.35_{\pm 2.22}$ | $72.76_{\pm 0.78}$ | $87.45_{\pm 0.25}$ | $80.78_{\pm 0.62}$ | $92.08_{\pm 1.61}$ | $87.97_{\pm 0.77}$ | $93.36_{\pm 0.56}$ | $\mathbf{95.85_{\pm 0.23}}$ | $72.19_{\pm 0.18}$ | $\mathbf{30.70_{\pm 1.21}}$ | $40.70_{\pm 2.30}$ | $40.29_{\pm 4.68}$ | 3.42 |
| $\text{AGG}_B$ (single) | $84.28_{\pm 1.57}$ | $72.78_{\pm 0.52}$ | $\mathbf{87.58_{\pm 0.38}}$ | $80.70_{\pm 0.00}$ | $92.00_{\pm 1.51}$ | $88.50_{\pm 0.73}$ | $92.01_{\pm 0.37}$ | $95.23_{\pm 0.27}$ | $72.09_{\pm 0.11}$ | $30.13_{\pm 0.73}$ | $41.91_{\pm 2.55}$ | $40.64_{\pm 4.83}$ | 3.33 |
| $\text{AGG}_B$ (ours) | $\mathbf{84.84_{\pm 1.39}}$ | $\mathbf{73.32_{\pm 0.85}}$ | $87.56_{\pm 0.27}$ | $80.75_{\pm 0.42}$ | $\mathbf{92.44_{\pm 1.42}}$ | $88.76_{\pm 0.65}$ | $\mathbf{93.54_{\pm 0.37}}$ | $95.79_{\pm 0.17}$ | $\mathbf{72.43_{\pm 0.16}}$ | $30.56_{\pm 0.84}$ | $\mathbf{42.39_{\pm 2.19}}$ | $\mathbf{40.96_{\pm 4.83}}$ | 1.58 |

## G. Extensive Ablation Study of Alternative Layer Architectures

In this section, we extend the results presented in Table 4 to all 12 datasets used in our experiments. As shown in Table 11, although several alternative layer architectures provide performance gains in specific cases under our training scheme and loss, only our original $\text{AGG}_B$ design consistently and significantly improves performance across all datasets.

We also evaluate a simplified, single-layer variant of $\text{AGG}_B$ that restricts the usable representation to only the immediate previous layer, $\boldsymbol{H}^{(l-1)}$, and is formulated as:

$$(\boldsymbol{D} + \boldsymbol{I})^{-1} \boldsymbol{H}^{(l-1)} \boldsymbol{W}^{(l)}.$$

This variant satisfies the same theoretical conditions—C1 (edge-awareness) and C2 (stability)—outlined in Section 4.2, just like our proposed architecture. While it improves performance on 10 out of 12 datasets—making it the most competitive alternative—the improvements are relatively marginal compared to those achieved by $\text{AGG}_B$.

The motivation for integrating representations from all preceding layers, rather than relying on a single layer, is to mitigate the risk of accumulating structural discrepancies. Ideally, $\text{AGG}_B$ could fully correct such inconsistencies at each layer. In practice, however, residual discrepancies may persist in intermediate layers and propagate through the network, ultimately affecting the final output. Relying solely on $\boldsymbol{H}^{(l-1)}$ risks amplifying these unresolved issues, whereas aggregating $\boldsymbol{H}^{(0:l-1)}$ enables $\text{AGG}_B$ to leverage earlier, potentially less corrupted representations, leading to more robust corrections.

Importantly, the performance gains from $\text{AGG}_B$ are not solely attributed to multi-layer integration. We also compare it with a JKNet-style block, which similarly aggregates outputs from all previous layers and is formulated as $\boldsymbol{H}^{(0:l-1)} \boldsymbol{W}^{(l)}$. $\text{AGG}_B$ outperforms this JKNet-style design on 9 out of 12 datasets, while the JKNet-style variant even degrades performance on 4 datasets. This result suggests that the inclusion of degree normalization, $(\boldsymbol{D} + \boldsymbol{I})^{-1}$—a key component of $\text{AGG}_B$ that ensures satisfaction of the conditions outlined in Section 4.2 (i.e., (1) edge-awareness and (2) stability)—is crucial for achieving consistent performance improvements across diverse datasets.

Although $\text{AGG}_B$ performs robustly in our experiments and ablations, we acknowledge that concatenating all preceding representations can introduce information redundancy and noise, particularly as GNN depth increases. However, this is not currently a critical issue, as GNNs typically achieve optimal performance at relatively shallow depths (e.g., two layers) due to over-smoothing. That said, as deeper GNNs become more effective in future research, developing more streamlined integration mechanisms that reduce redundancy and noise presents a promising direction for extending this work.

## H. Extensive Ablation Study on Deeper GCNs

In this section, we evaluate the effectiveness of $\text{AGG}_B$ when applied to deeper GCN, using 4, 8, 12, 16, and 20-layer models across 6 datasets. In addition to the standard GCN and $\text{GCN}_B$, we include two more variants: (1) GCN trained with DropEdge, and (2) $\text{DropEdge}_B$, where $\text{AGG}_B$ is integrated into a GCN trained with DropEdge.

These variants are included to assess whether $\text{AGG}_B$ can provide further improvements beyond what DropEdge achieves, particularly in deep architectures. This is motivated by the original DropEdge paper (Rong et al., 2020), which highlights its effectiveness in alleviating oversmoothing and demonstrates more substantial performance gains in deeper GNNs. The results are presented in Table 12.

$\text{AGG}_B$ improves performance in 28 out of 30 configurations, demonstrating that its effectiveness is robust to architectural depth. Notably, performance gains tend to increase with depth, suggesting that deeper GNNs are more susceptible

*Table 12.* Accuracy with varying GCN depths. DropEdge$_B$ denotes a GCN trained with DropEdge followed by integration of AGG$_B$. All experiments use a hidden dimension of 256 and a learning rate of 0.001. For fair comparison, the edge dropping ratio is fixed at 0.5 for both DropEdge and AGG$_B$, and the hyperparameter $\lambda$ is fixed at 1.0. Integrating AGG$_B$ improves performance in 56 out of 60 configurations.

| | Cora | | | | Pubmed | | | | A.Computer | | | |
|---|---|---|---|---|---|---|---|---|---|---|---|---|
| | GCN | GCN$_B$ | DropEdge | DropEdge$_B$ | GCN | GCN$_B$ | DropEdge | DropEdge$_B$ | GCN | GCN$_B$ | DropEdge | DropEdge$_B$ |
| 4 | $81.98_{\pm1.45}$ | $\mathbf{82.50_{\pm1.65}}$ | $82.51_{\pm1.76}$ | $\mathbf{82.96_{\pm1.83}}$ | $83.73_{\pm0.20}$ | $\mathbf{86.87_{\pm0.55}}$ | $84.13_{\pm0.25}$ | $\mathbf{87.14_{\pm0.35}}$ | $86.66_{\pm0.53}$ | $\mathbf{86.82_{\pm0.49}}$ | $86.23_{\pm1.11}$ | $\mathbf{86.34_{\pm1.14}}$ |
| 8 | $77.54_{\pm2.05}$ | $\mathbf{79.99_{\pm1.45}}$ | $79.64_{\pm1.89}$ | $\mathbf{80.86_{\pm1.44}}$ | $83.07_{\pm0.16}$ | $\mathbf{85.90_{\pm0.48}}$ | $83.19_{\pm0.27}$ | $\mathbf{85.49_{\pm0.58}}$ | $80.60_{\pm2.22}$ | $\mathbf{81.07_{\pm2.33}}$ | $79.78_{\pm1.86}$ | $\mathbf{80.21_{\pm1.92}}$ |
| 12 | $73.42_{\pm2.08}$ | $\mathbf{77.70_{\pm1.99}}$ | $78.41_{\pm1.71}$ | $\mathbf{79.65_{\pm2.01}}$ | $82.44_{\pm0.29}$ | $\mathbf{85.48_{\pm0.68}}$ | $82.53_{\pm0.35}$ | $\mathbf{84.64_{\pm0.52}}$ | $75.08_{\pm2.24}$ | $\mathbf{76.45_{\pm2.06}}$ | $76.17_{\pm3.65}$ | $\mathbf{77.16_{\pm3.75}}$ |
| 16 | $69.78_{\pm3.45}$ | $\mathbf{75.67_{\pm3.32}}$ | $77.92_{\pm1.92}$ | $\mathbf{79.39_{\pm1.72}}$ | $82.08_{\pm0.45}$ | $\mathbf{85.21_{\pm0.64}}$ | $81.76_{\pm0.47}$ | $\mathbf{83.78_{\pm0.68}}$ | $73.50_{\pm4.52}$ | $\mathbf{74.83_{\pm4.73}}$ | $74.89_{\pm1.42}$ | $\mathbf{76.20_{\pm1.41}}$ |
| 20 | $64.15_{\pm5.08}$ | $\mathbf{72.23_{\pm3.45}}$ | $73.89_{\pm2.99}$ | $\mathbf{76.31_{\pm3.41}}$ | $81.75_{\pm0.55}$ | $\mathbf{85.23_{\pm0.75}}$ | $81.17_{\pm0.36}$ | $\mathbf{83.20_{\pm0.51}}$ | $67.99_{\pm3.98}$ | $\mathbf{68.83_{\pm4.42}}$ | $72.14_{\pm2.88}$ | $\mathbf{73.69_{\pm3.05}}$ |

| | CS | | | | Squirrel | | | | Chameleon | | | |
|---|---|---|---|---|---|---|---|---|---|---|---|---|
| | GCN | GCN$_B$ | DropEdge | DropEdge$_B$ | GCN | GCN$_B$ | DropEdge | DropEdge$_B$ | GCN | GCN$_B$ | DropEdge | DropEdge$_B$ |
| 4 | $89.63_{\pm0.40}$ | $\mathbf{91.61_{\pm0.34}}$ | $89.70_{\pm0.37}$ | $\mathbf{91.51_{\pm0.43}}$ | $33.96_{\pm1.01}$ | $\mathbf{34.37_{\pm1.31}}$ | $33.76_{\pm1.42}$ | $\mathbf{33.82_{\pm1.38}}$ | $39.46_{\pm3.17}$ | $\mathbf{40.00_{\pm3.05}}$ | $37.95_{\pm4.27}$ | $\mathbf{38.96_{\pm4.17}}$ |
| 8 | $88.44_{\pm0.38}$ | $\mathbf{91.33_{\pm0.32}}$ | $88.58_{\pm0.22}$ | $\mathbf{91.24_{\pm0.18}}$ | $34.06_{\pm1.28}$ | $\mathbf{34.41_{\pm1.96}}$ | $34.14_{\pm1.47}$ | $\mathbf{34.41_{\pm1.38}}$ | $37.55_{\pm2.51}$ | $37.33_{\pm2.86}$ | $\mathbf{37.99_{\pm3.98}}$ | $37.06_{\pm4.31}$ |
| 12 | $86.84_{\pm0.48}$ | $\mathbf{90.57_{\pm0.28}}$ | $87.89_{\pm0.26}$ | $\mathbf{91.03_{\pm0.33}}$ | $\mathbf{34.78_{\pm1.67}}$ | $34.74_{\pm1.69}$ | $34.91_{\pm1.74}$ | $\mathbf{35.05_{\pm1.65}}$ | $37.18_{\pm4.83}$ | $\mathbf{37.32_{\pm5.14}}$ | $36.11_{\pm4.43}$ | $35.25_{\pm3.76}$ |
| 16 | $85.04_{\pm0.81}$ | $\mathbf{89.98_{\pm0.52}}$ | $87.41_{\pm0.44}$ | $\mathbf{90.76_{\pm0.53}}$ | $34.49_{\pm1.66}$ | $\mathbf{34.70_{\pm1.96}}$ | $34.77_{\pm1.39}$ | $\mathbf{34.90_{\pm1.28}}$ | $36.35_{\pm3.37}$ | $\mathbf{36.59_{\pm2.50}}$ | $34.92_{\pm5.12}$ | $\mathbf{35.88_{\pm3.57}}$ |
| 20 | $82.34_{\pm1.86}$ | $\mathbf{88.59_{\pm1.47}}$ | $85.55_{\pm0.99}$ | $\mathbf{88.77_{\pm1.08}}$ | $34.23_{\pm1.24}$ | $\mathbf{34.80_{\pm1.61}}$ | $34.83_{\pm1.33}$ | $\mathbf{34.92_{\pm1.58}}$ | $35.63_{\pm4.37}$ | $37.30_{\pm4.52}$ | $34.57_{\pm4.63}$ | $34.66_{\pm3.92}$ |

*Table 13.* Accuracy from 5 independent runs of a 2-layer GCN (hidden dimension: 256), and after integration of AGG$_B$ (GCN$_B$), evaluated on the public split of larger datasets: Flickr, Ogbn-arxiv, and Reddit. The GCN is trained using fixed hyperparameters (learning rate: 0.001, dropout: 0.5), while AGG$_B$ is trained with fixed parameters ($\lambda = 1.0$, DropEdge rate: 0.5). Integrating AGG$_B$ consistently improves overall performance, with the largest gains observed in low-degree, heterophilic nodes.

| | Flickr | | Arxiv | | Reddit | |
|---|---|---|---|---|---|---|
| | GCN | GCN$_B$ | GCN | GCN$_B$ | GCN | GCN$_B$ |
| **Overall Accuracy** | $52.50_{\pm0.15}$ | $\mathbf{52.84_{\pm0.08}}$ | $71.06_{\pm0.10}$ | $\mathbf{71.37_{\pm0.10}}$ | $94.61_{\pm0.01}$ | $\mathbf{94.89_{\pm0.01}}$ |
| High-degree Nodes | $49.66_{\pm0.26}$ | $\mathbf{49.87_{\pm0.18}}$ | $\mathbf{80.06_{\pm0.11}}$ | $79.95_{\pm0.14}$ | $98.81_{\pm0.01}$ | $\mathbf{98.84_{\pm0.01}}$ |
| Low-degree Nodes | $53.93_{\pm0.28}$ | $\mathbf{54.58_{\pm0.16}}$ | $62.32_{\pm0.05}$ | $\mathbf{63.16_{\pm0.04}}$ | $88.06_{\pm0.01}$ | $\mathbf{88.84_{\pm0.03}}$ |
| Homophilous Nodes | $\mathbf{80.58_{\pm0.49}}$ | $80.44_{\pm0.10}$ | $\mathbf{94.87_{\pm0.02}}$ | $94.60_{\pm0.03}$ | $\mathbf{99.74_{\pm0.01}}$ | $99.66_{\pm0.01}$ |
| Heterophilous Nodes | $18.00_{\pm0.07}$ | $\mathbf{18.18_{\pm0.07}}$ | $32.30_{\pm0.09}$ | $\mathbf{33.78_{\pm0.08}}$ | $84.10_{\pm0.01}$ | $\mathbf{85.01_{\pm0.02}}$ |

to structural inconsistencies as representations undergo repeated aggregation—thus creating greater opportunities for improvement via enhanced edge-robustness.

As expected, DropEdge yields more substantial improvements in deeper architectures, while its effects remain marginal in shallow ones. Importantly, integrating AGG$_B$ into DropEdge-trained models significantly boosts performance in 28 out of 30 settings. This demonstrates that AGG$_B$ provides a distinct benefit—specifically, enhanced edge-robustness. These results reinforce our claim that DropEdge alone is insufficient for addressing edge-robustness, regardless of model depth, and that AGG$_B$ offers a principled approach to mitigating structural inconsistencies in deep GNNs.

## I. Additional Experiments on Larger Datasets

To further demonstrate the broad applicability of AGG$_B$, we include results on three larger datasets: Arxiv (Hu et al., 2020a), Reddit (Hamilton et al., 2017), and Flickr (Zeng et al., 2020), all of which are loaded from the Deep Graph Library (DGL). As shown in Table 13, AGG$_B$ consistently improves performance across all three datasets, in line with earlier findings. In all cases, the performance gains primarily stem from improvements on low-degree and heterophilous nodes, highlighting that the observed benefits are indeed driven by enhanced edge-robustness. It is also worth noting that these results are obtained without any hyperparameter tuning. This suggests that further improvements are possible with tuning—as the larger performance gain observed on Arxiv in Table 1.

Additionally, we conduct the edge removal experiments described in Appendix F. The performance degradation from random edge removal is significantly reduced when using AGG$_B$, further validating its effectiveness on larger-scale datasets.

*Table 14.* Accuracy under random edge removal (%) in test, using the same experimental settings as Table 13. $\text{AGG}_B$ consistently improves performance across all edge removal ratios and datasets, with greater gains at higher removal ratios.

| | Flickr | | Arxiv | | Reddit | |
|---|---|---|---|---|---|---|
| | GCN | $\text{GCN}_B$ | GCN | $\text{GCN}_B$ | GCN | $\text{GCN}_B$ |
| 100% | $52.50_{\pm 0.15}$ | $\mathbf{52.84_{\pm 0.08}}$ | $71.06_{\pm 0.10}$ | $\mathbf{71.37_{\pm 0.10}}$ | $94.61_{\pm 0.01}$ | $\mathbf{94.89_{\pm 0.01}}$ |
| 75% | $50.95_{\pm 0.12}$ | $\mathbf{51.56_{\pm 0.11}}$ | $69.58_{\pm 0.08}$ | $\mathbf{70.59_{\pm 0.07}}$ | $94.47_{\pm 0.01}$ | $\mathbf{94.87_{\pm 0.01}}$ |
| 50% | $47.49_{\pm 0.52}$ | $\mathbf{49.66_{\pm 0.21}}$ | $67.44_{\pm 0.11}$ | $\mathbf{69.00_{\pm 0.06}}$ | $94.18_{\pm 0.01}$ | $\mathbf{94.82_{\pm 0.01}}$ |
| 25% | $41.31_{\pm 1.06}$ | $\mathbf{45.82_{\pm 0.34}}$ | $62.50_{\pm 0.08}$ | $\mathbf{65.50_{\pm 0.06}}$ | $93.52_{\pm 0.01}$ | $\mathbf{94.38_{\pm 0.03}}$ |
| 0% | $31.73_{\pm 1.33}$ | $\mathbf{39.57_{\pm 0.70}}$ | $45.36_{\pm 0.19}$ | $\mathbf{54.56_{\pm 0.04}}$ | $38.91_{\pm 0.01}$ | $\mathbf{45.48_{\pm 0.27}}$ |

## J. Generalizing Theorem 3.9 to Other GNN Architectures

In this section, we extend our discrepancy analysis—Theorem 3.9—beyond GCN to a broader class of GNN architectures. We provide proofs for three representative models: GraphSAGE, GIN, and GAT, which are also used in our experiments to assess the generalizability of $\text{AGG}_B$, as presented in Section 6.4. These results theoretically demonstrate that the issue of non-optimizable edge-robustness is not specific to GCN, but is a fundamental limitation shared across various GNN architectures—one that $\text{AGG}_B$ is designed to address. We omit SGC from this analysis, as it can be regarded as a linearized variant of GCN and is therefore already covered by the proof in Appendix C.

### J.1. GraphSAGE

The GraphSAGE layer is formulated as:

$$\boldsymbol{H}^{(l)} = \sigma(\text{AGG}^{(l)}(\boldsymbol{H}^{(l-1)}, \boldsymbol{A}) + \boldsymbol{H}^{(l-1)}\boldsymbol{W}_2^{(l)}) = \sigma(\hat{\boldsymbol{A}}\boldsymbol{H}^{(l-1)}\boldsymbol{W}_1^{(l)} + \boldsymbol{H}^{(l-1)}\boldsymbol{W}_2^{(l)}),$$

where $\hat{\boldsymbol{A}} = \boldsymbol{D}^{-1}\boldsymbol{A}$ denotes the normalized adjacency matrix. Then, the discrepancy at layer $l$ satisfies:

$$
\begin{aligned}
\|\boldsymbol{H}_1^{(l)} - \boldsymbol{H}_2^{(l)}\|_2 &\leq L_\sigma \|\text{AGG}^{(l)}(\boldsymbol{H}_1^{(l-1)}, \boldsymbol{A}_1) - \text{AGG}^{(l)}(\boldsymbol{H}_2^{(l-1)}, \boldsymbol{A}_2) + \boldsymbol{H}_1^{(l-1)}\boldsymbol{W}_2^{(l)} - \boldsymbol{H}_2^{(l-1)}\boldsymbol{W}_2^{(l)}\|_2 \\
&\leq L_\sigma \|\hat{\boldsymbol{A}}_1 \boldsymbol{H}_1^{(l-1)}\boldsymbol{W}_1^{(l)} - \hat{\boldsymbol{A}}_2 \boldsymbol{H}_2^{(l-1)}\boldsymbol{W}_1^{(l)}\|_2 + L_\sigma \|\boldsymbol{W}_2^{(l)}\|_2 \|\boldsymbol{H}_1^{(l-1)} - \boldsymbol{H}_2^{(l-1)}\|_2 \\
&\leq L_\sigma (\|\boldsymbol{W}_1^{(l)}\|_2 + \|\boldsymbol{W}_2^{(l)}\|_2) \|\boldsymbol{H}_1^{(l-1)} - \boldsymbol{H}_2^{(l-1)}\|_2 + L_\sigma |V| \|\boldsymbol{W}_1^{(l)}\|_2 \|\hat{\boldsymbol{A}}_1 - \hat{\boldsymbol{A}}_2\|_2 \\
&\leq C_1 \|\boldsymbol{H}_1^{(l-1)} - \boldsymbol{H}_2^{(l-1)}\|_2 + C_2
\end{aligned}
$$

where $C_1 = L_\sigma(\|\boldsymbol{W}_1^{(l)}\|_2 + \|\boldsymbol{W}_2^{(l)}\|_2), C_2 = L_\sigma |V| \|\boldsymbol{W}_1^{(l)}\|_2 \|\hat{\boldsymbol{A}}_1 - \hat{\boldsymbol{A}}_2\|_2$.

### J.2. GIN

The GIN layer is formulated as:

$$\boldsymbol{H}^{(l)} = \text{MLP}^{(l)}(\text{AGG}^{(l)}(\boldsymbol{H}^{(l-1)}, \boldsymbol{A}) + (1 + \epsilon^{(l)})\boldsymbol{H}^{(l-1)}) = \text{MLP}^{(l)}(\boldsymbol{A}\boldsymbol{H}^{(l-1)} + (1 + \epsilon^{(l)})\boldsymbol{H}^{(l-1)}),$$

where $\epsilon^{(l)}$ is a learnable scalar at layer $l$. Then, the discrepancy at layer $l$ satisfies:

$$
\begin{aligned}
\|\boldsymbol{H}_1^{(l)} - \boldsymbol{H}_2^{(l)}\|_2 &\leq C \|\text{AGG}^{(l)}(\boldsymbol{H}_1^{(l-1)}, \boldsymbol{A}_1) - \text{AGG}^{(l)}(\boldsymbol{H}_2^{(l-1)}, \boldsymbol{A}_2) + (1 + \epsilon^{(l)})\boldsymbol{H}_1^{(l-1)} - (1 + \epsilon^{(l)})\boldsymbol{H}_2^{(l-1)}\|_2 \\
&\leq C \|\boldsymbol{A}_1 \boldsymbol{H}_1^{(l-1)} - \boldsymbol{A}_2 \boldsymbol{H}_2^{(l-1)}\|_2 + C|1 + \epsilon^{(l)}| \|\boldsymbol{H}_1^{(l-1)} - \boldsymbol{H}_2^{(l-1)}\|_2 \\
&\leq C(\|\boldsymbol{A}_1\|_2 + |1 + \epsilon^{(l)}|) \|\boldsymbol{H}_1^{(l-1)} - \boldsymbol{H}_2^{(l-1)}\|_2 + C|V| \|\boldsymbol{A}_1 - \boldsymbol{A}_2\|_2 \\
&\leq C_1 \|\boldsymbol{H}_1^{(l-1)} - \boldsymbol{H}_2^{(l-1)}\|_2 + C_2
\end{aligned}
$$

where $C$ is the discrepancy bound of $\text{MLP}^{(l)}$, $C_1 = C(\|\boldsymbol{A}_1\|_2 + |1 + \epsilon^{(l)}|), C_2 = C|V| \|\boldsymbol{A}_1 - \boldsymbol{A}_2\|_2$.

## J.3. GAT

The GAT layer is defined as:

$$\mathbf{h}_i^{(l)} = \sigma\left(\sum_{j \in \mathcal{N}_i} \alpha_{ij} \mathbf{W}_1^{(l)} \mathbf{h}_j^{(l-1)}\right),$$

$$\alpha_{ij} = \frac{\exp(\text{LeakyReLU}(\mathbf{a}^\top[\mathbf{W}'\mathbf{h}_i^{(l-1)} \| \mathbf{W}'\mathbf{h}_j^{(l-1)}]))}{\sum_{k \in \mathcal{N}_i} \exp(\text{LeakyReLU}(\mathbf{a}^\top[\mathbf{W}'\mathbf{h}_i^{(l-1)} \| \mathbf{W}'\mathbf{h}_k^{(l-1)}]))}$$

$$= \frac{\exp(\text{LeakyReLU}(\mathbf{p}^\top \mathbf{W}'\mathbf{h}_i^{(l-1)} + \mathbf{q}^\top \mathbf{W}'\mathbf{h}_j^{(l-1)}))}{\sum_{k \in \mathcal{N}_i} \exp(\text{LeakyReLU}(\mathbf{p}^\top \mathbf{W}'\mathbf{h}_i^{(l-1)} + \mathbf{q}^\top \mathbf{W}'\mathbf{h}_k^{(l-1)})))},$$

where $\mathbf{a} \in \mathbb{R}^{2F'}$ is the original attention weight vector, and $\mathbf{p}, \mathbf{q} \in \mathbb{R}^{F'}$ are its components such that $\mathbf{a}^\top[\cdot\|\cdot] = \mathbf{p}^\top(\cdot) + \mathbf{q}^\top(\cdot)$. The induced attention matrix can be interpreted as:

$$\boldsymbol{A}^* = \text{RowNorm}(\exp(\text{LeakyReLU}(\text{diag}(\mathbf{p}^\top \mathbf{W}'\mathbf{H}^{(l-1)^\top})\mathbf{A} + \mathbf{A}\text{diag}(\mathbf{q}^\top \mathbf{W}'\mathbf{H}^{(l-1)^\top})))).$$

Since $\boldsymbol{A}^*$ is row-stochastic, we have $\|\boldsymbol{A}^*\|_2 = 1$. The discrepancy is thus bounded by:

$$\|\boldsymbol{H}_1^{(l)} - \boldsymbol{H}_2^{(l)}\|_2 \leq L_\sigma \|\text{AGG}^{(l)}(\boldsymbol{H}_1^{(l-1)}, \boldsymbol{A}_1) - \text{AGG}^{(l)}(\boldsymbol{H}_2^{(l-1)}, \boldsymbol{A}_2)\|_2$$

$$\leq L_\sigma \|\boldsymbol{A}_1^* \boldsymbol{H}_1^{(l-1)} \boldsymbol{W}_1^{(l)} - \boldsymbol{A}_2^* \boldsymbol{H}_2^{(l-1)} \boldsymbol{W}^{(l)}\|_2$$

$$\leq L_\sigma \|\boldsymbol{W}_1^{(l)}\| \|\boldsymbol{A}_1^* \boldsymbol{H}_1^{(l-1)} - \boldsymbol{A}_1^* \boldsymbol{H}_2^{(l-1)} + \boldsymbol{A}_1^* \boldsymbol{H}_2^{(l-1)} - \boldsymbol{A}_2^* \boldsymbol{H}_2^{(l-1)}\|_2$$

$$\leq L_\sigma \|\boldsymbol{W}_1^{(l)}\| \|\boldsymbol{H}_1^{(l-1)} - \boldsymbol{H}_2^{(l-1)}\|_2 + L_\sigma \|\boldsymbol{W}_1^{(l)}\|_2 \|\boldsymbol{H}_2^{(l-1)}\|_2 \|\boldsymbol{A}_1^* - \boldsymbol{A}_2^*\|_2$$

$$\leq C_1 \|\boldsymbol{H}_1^{(l-1)} - \boldsymbol{H}_2^{(l-1)}\|_2 + C_2,$$

where $C_1 = L_\sigma \|\boldsymbol{W}_1^{(l)}\|_2, C_2 = C_1 |V| \|\boldsymbol{A}_1^* - \boldsymbol{A}_2^*\|_2$.

