# OpenReview forum: "Aggregation Buffer: Revisiting DropEdge with a New Parameter Block"
_ICML.cc/2025/Conference — ICML 2025 poster_

### Official Review · Reviewer_Zx9t · 2025-03-04

**Overall Recommendation:** 3

**Summary:**

This paper revisit dropedge, it claims the robustness of GNNs will grow bad during training, which yields poor performance. This paper propose aggregation buffer, a block designed to address this problem.

## Update after rebuttal

I recommend accept as a poster.

**Claims And Evidence:**

Yes.

The main claim is that Dropedge is helpful for the robustness but harmful for the bias. The evidence of this claim is in Figure 2. And this claim is also intuitive and sounds correct to me (I actually know the fact Dropedge is helpful for the robustness from a paper called Dropmessage).

**Essential References Not Discussed:**

No

**Experimental Designs Or Analyses:**

Yes. No obvious issue.

Table 1 uses strong baselines, Table 2 uses some different encoders. The authors also include results of high-degree and low-degree nodes, I do not know why this is necessary, but it is harmless. The overall experiments are decent, at least in my batch.

**Methods And Evaluation Criteria:**

Yes.

The method is to add a new aggregation block that satisfied two conditions (actually one condition, since C2 implies C1). From Figure 2 we see this method works. But in theory, why these two conditions is enough for the bias-robustness trade-off is a bit unclear.

**Other Comments Or Suggestions:**

No.

**Other Strengths And Weaknesses:**

S1: The paper is clear and the analysis seems reasonable.

S2: The experiments are comprehensive.

W1: The improvements seem marginal.

W2: The paper aims to improve the robustness of GNNs, but the experiments are conducted on the original graph.

**Questions For Authors:**

Q1: There is a recent paper [1] that report high performance of classic GNNs, e.g., the raw GCN achieves 85.10 on Cora. Is there difference  between your setup and theirs?

[1]Classic GNNs are Strong Baselines: Reassessing GNNs for Node Classification

**Relation To Broader Scientific Literature:**

The paper is related to DropEdge, JKNet.

**Theoretical Claims:**

Yes. No obvious issue.

Theorem 4.1 basically says the proposed block satisfied the two condition, I have checked the correctness.

---

> ### Author Rebuttal · Authors · 2025-04-01
>
> We sincerely appreciate your detailed review and valuable questions. We hope the following responses address your concerns.
>
> **Why the two conditions are enough for the bias-robustness trade-off is unclear.**
>
> Thank you for raising this insightful point.
> Our two conditions and the layer-wise correction mechanism of AGG$_B$ are motivated by our discrepancy-bound analysis (Sec. 3.4).
> Condition C1 encourages AGG$_B$ to adapt to structural variations, while C2 ensures stability by minimizing unnecessary changes when the graph structure is stable.
> Together, these conditions mitigate representation inconsistencies caused by structural variations in a *layer-wise manner*.
>
> Establishing a rigorous theoretical link between layer-wise corrections and robustness—defined on the *final output of a GNN*—is indeed challenging.
> However, it is intuitive that consistent corrections at each layer yield a more robust final representation—a notion supported by our empirical results.
> Moreover, in the case of a single-layer GNN, the layer-wise and final-output views coincide, offering a clear example where this connection holds.
>
> **W1. The improvements seem marginal.**
>
> We believe that a difference in experimental setups should be considered when evaluating the significance of improvements.
> Our method aims to enhance trained GNNs—similar to curriculum learning (e.g., TUNEUP) or graph augmentation (e.g., GraphPatcher)—rather than training GNNs from scratch.
> While prior works often use fixed hyperparameters for base GNNs, we performed a grid search even for the base models, boosting the *base accuracy* to improve upon.
>
> Although this setup may yield smaller apparent gains, it more accurately reflects the true robustness and effectiveness of methods.
> Notably, our approach is the only method that consistently improves performance across all datasets, suggesting that these gains stem from architectural changes--the integration of AGG$_B$--rather than hyperparameter tuning.
> These results support our view that addressing structural inconsistency offers further opportunities to enhance GNN performance.
>
> **W2. The paper aims to improve the robustness of GNNs, but the experiments are conducted on the original graph.**
>
> Thank you for your helpful suggestion.
> We focused on original graphs because our target is robustness among nodes with different structural properties within the same graph, rather than across graphs.
> To evaluate this, we measured performance across groups based on node degrees (low vs. high) and structural roles (heterophilic vs. homophilic), which are closely linked to structural inconsistency.
> Our method showed significant gains for low-degree and heterophilic nodes, supporting the claim that AGG$_B$ improves edge robustness.
>
> Nevertheless, we agree that testing under graph perturbation is informative.
> In response to your feedback, we conducted additional experiments using random edge removal in test graphs ([link - Table C](https://shorturl.at/GjlH9)).
> AGG$_B$ significantly improves the standard GCN—even those trained with DropEdge—demonstrating its robustness benefits.
> Furthermore, GNNs trained with DropEdge did not retain performance any better than those without it, reinforcing our claim that DropEdge alone is insufficient for robustness due to inherent inductive bias.
>
> **Q1: There is a recent paper [1] that report high performance of classic GNNs, (e.g. 85.10 on Cora). Is there difference between your setup and theirs?**
>
> We carefully reviewed [1] and identified several key differences in experimental setups.
> In our work, we use 10 random splits per dataset, running one trial per split.
> We select hyperparameters based on validation results from 5 of these splits to avoid overfitting to a specific partition.
>
> In contrast, [1] employs fixed public splits—using a single public split for Cora—with 5 runs using random weight initializations.
> This approach may yield higher accuracy and lower variance by tuning specifically for that split, but it might not generalize well to other partitions.
> This approach may yield higher accuracy and lower variance due to tuning specifically for that split, but it might not generalize well to other data partitions.
>
> Additionally, their hyperparameter search space is broader, including batch/layer normalization, residual connections, number of layers, additional linear transformations before and after GNN layers, and maximum epochs.
> Our search includes dropout, hidden dimension, learning rate, and weight decay, following standard practices in previous GNN literature [2]. These differences likely explain the performance gap.
>
> Thank you again for your insightful review and for considering our response.
>
> [1] Classic gnns are strong baselines: Reassessing gnns for node classification.
>
> [2] Pitfalls of Graph Neural Network Evaluation

---

### Official Review · Reviewer_Yzw2 · 2025-03-11

**Overall Recommendation:** 3

**Summary:**

This paper analyzes the robustness of GNN under dropping edges and proposes Aggregation Buffer (AGGB) as a solution, which enhances the robustness of GNN through a two-step training strategy while maintaining the knowledge of the original model. AGGB optimizes the shortcomings of DropEdge and improves the performance of GNN under unstable graph structures.

**Claims And Evidence:**

The main claim of the paper is that GNN cannot effectively cope with the adjacency matrix perturbations caused by DropEdge due to its aggregation operation , resulting in performance degradation. To this end, the authors proposed AGGB, a module that can enhance the robustness of GNN, aiming to solve this problem. In the experimental section, the authors verified the effectiveness of AGGB through a series of comparative experiments. The experimental results provide evidence to support the claims in the paper.

**Essential References Not Discussed:**

No

**Experimental Designs Or Analyses:**

The experimental datasets are common datasets for semi-supervised learning. The evaluation method of degree bias and structural disparity is one of the key innovations of the experimental design in the paper. By evaluating the performance of the head node and the tail node respectively, the robustness of the model under different degree distributions can be better measured. For structural differences, nodes are divided into homogeneous nodes and heterogeneous nodes based on their homogeneity ratio. This is also an innovation.

**Methods And Evaluation Criteria:**

The proposed method mainly includes Aggregation Buffer (AGGB), which is used to enhance the robustness of GNN under structural perturbations. The proposed evaluation criteria is appropriate.

**Other Comments Or Suggestions:**

See the questions.

**Other Strengths And Weaknesses:**

Strengths:

1. The proposed method is a novel approach for improving the robustness of GNNs under graph perturbations, as it addresses an important issue of structural changes in graph data.

2. The paper does a good job of clearly explaining the theories behind the designation and shows why existing methods fail to address these issues.
3. The experimental design is solid, with a wide range of datasets and a novel evaluation framework, including tests for robustness under GNNs.

Weaknesses:

​There are no major weakness in this paper.

**Questions For Authors:**

1. If there is still a large deviation between Q and the true distribution P, is it reasonable to assume that Q is sufficiently close to P?
2. In the current design, AGGB relies on the representations of all the first 𝑙 layers and the adjacency matrix. Does this design lead to information redundancy or introduce unnecessary noise? Is there a more streamlined way to integrate this information?
3. Currently, the method trains GNN first and then AGGB. Is it possible to introduce some AGGB mechanisms when pre-training GNN?

**Relation To Broader Scientific Literature:**

GNN is widely used in various graph data tasks. DropEdge (Rong et al., 2019) proposed to enhance the robustness of the model by randomly deleting edges in the graph. However, this method only trains the model by perturbing the graph structure, and does not fundamentally solve the sensitivity of the GNN structure to perturbations. Based on this, this paper further proposes to introduce the AGGB module to correct the performance of GNN under perturbations, so that the model can adapt to different graph structure changes and improve robustness.

**Theoretical Claims:**

In the paper, several key theoretical claims are proposed. I checked the claims. In general, the theoretical claims are effectively supported by mathematical derivation and theoretical proof.

---

> ### Author Rebuttal · Authors · 2025-04-01
>
> We sincerely appreciate your detailed review and valuable questions. We have reordered your questions since we believe Q1 and Q3 are closely related. We hope the following responses adequately address your concerns.
>
> **Q2. Does the current design of AGG$_B$ which relies on all preceding representations lead to information redundancy or introduce unnecessary noise? Is there a more streamlined way to integrate this information?**
>
> Thank you for raising this point.
> There can be information redundancy in our design, but our rationale for integrating representations from all previous layers is to minimize the propagation of structural discrepancies.
> Ideally, AGG$_B$ would fully correct inconsistencies at each layer; however, in practice, some discrepancies may remain unresolved in intermediate layers and can accumulate in deeper ones.
> Relying solely on the immediately preceding layer $H^{(l-1)}$ risks carrying these issues forward, whereas leveraging all prior representations $H^{(0;l-1)}$ allows AGG$_B$ to access earlier, less corrupted information for more robust corrections.
>
> For detailed analysis, we have strengthened our ablation study ([link - Table A, B](https://shorturl.at/2cblM)) (a) by including a single-layer variant $(D+I)^{-1}H^{l-1}W^{l}$ of AGG$_B$ and (b) running it for all datasets (beyond the 4 originally used).
> Although this variant improves GNN performance in most cases, our original design consistently yields stronger results across all datasets and achieves the highest overall ranking.
> Developing a more streamlined integration that minimizes information redundancy and noise is a promising future direction of this work.
>
> **Q1. If there is still a large deviation between Q and the true distribution P, is it reasonable to assume that Q is sufficiently close to P?**
>
> Thank you for this insightful question.
> The assumption $Q \approx P$ is used in our approximation between Eq. (3) and Eq. (4):
> $$
> E_{P}[ \log Q(y_i |G_i) - \log Q(y_i|\tilde{G_i})] \approx D_{KL} (Q(y_i |G_i) \Vert Q(y_i|\tilde{G_i}))
> $$
> This approximation is adopted because the true distribution P is inaccessible.
> We agree that assuming $Q \approx P$ is generally not valid when Q significantly deviates from P.
> However, as the training proceeds with the bias term $D_{KL}(P(y_i|G_i) \Vert Q(y_i|G_i))$ optimized, it brings $Q$ closer to $P$, at least on the training distribution.
> Furthermore, our two-step training scheme leverages this assumption in loss function only after base GCN is trained, which makes the assumption more reliable in practice.
>
> To support the reliability of our framework, we conducted an experiment to indirectly assess the true distribution $P$ using the labels by the below formulation.
> $$
> \frac{1}{N} \sum_{i=1}^N \sum_{c=1}^{C} y_i(c) (\log Q(c|G_i) - \log Q(c|\tilde{G}_i))
> $$
> Although this expression uses one-hot ground truth labels (thus only sampling one class per node) and the labels include noise, it still offers a useful proxy to assess the validity of the approximation.
>
> Our results ([link - Figure A](https://shorturl.at/MeZ73)) show that its shape closely mirrors that of the approximated KL divergence, indicating that our approximation captures trends well.
> Furthermore, AGG$_B$ shows robust optimization behavior even under this label-based approximation, demonstrating its effectiveness in terms of robustness optimization.
> We will include this discussion in the final version.
>
> **Q3. Currently, the method trains GNN first and then $AGG_B$. Is it possible to introduce a mechanism for training $AGG_B$ when pre-training GNN?**
>
> This is an insightful question that relates closely to the assumption discussed above.
> The assumption $P \approx Q$ is used once more in the bias term of Eq. (5) due to inaccessibility to true distribution.
> $$
> D_{KL} (P(y_i |G_i) \Vert Q_B(y_i|G_i)) \approx D_{KL} (Q(y_i |G_i) \Vert Q_B(y_i|G_i))
> $$
> In our two-step scheme, the pre-trained $Q$ is optimized to be close to $P$, making this approximation more valid, especially when restricted to training samples (Sec. 4.3)
> However, if AGG$_B$ were trained jointly with the GNN from scratch, $Q$ would initially deviate significantly from $P$, and the robustness-controlled loss could interfere with the optimization of the standard loss, leading to suboptimal guidance.
>
> Nonetheless, we agree that incorporating AGG$_B$ into the end-to-end training of GNNs would enhance the applicability and elegance of our approach.
> As noted in our conclusion, this is a promising direction for future work, and we are actively exploring strategies to integrate it into joint training.
>
> Thank you again for your insightful reviews and for taking the time to read our response.

---

### Official Review · Reviewer_jKJV · 2025-03-12

**Overall Recommendation:** 1

**Summary:**

This paper revisits DropEdge, a data augmentation technique for GNNs that randomly removes edges to enhance robustness. While DropEdge helps mitigate overfitting, its performance gains in supervised learning are limited due to an inherent inductive bias in GNN architectures. To address this, the authors propose Aggregation Buffer ($AGG_B$), a parameter block that improves GNN robustness and enhances DropEdge’s effectiveness. $AGG_B$ can be integrated as a post-processing step in any GNN model. Empirical results on 11 node classification benchmarks show that $AGG_B$ significantly improves accuracy and mitigates degree bias and structural disparity. The paper provides a theoretical analysis of DropEdge’s limitations and demonstrates that $AGG_B$ serves as a unifying solution to structural inconsistencies in graph data.

**Claims And Evidence:**

The proposed \( AGG_B \) takes \( H^{(0:l-1)} \) as input, rather than the standard 1-hop neighborhood representation \( H^{l-1} \). It is well known that incorporating \( H^{(0:l-1)} \) can enhance performance, as demonstrated in works like JKNet. However, this raises an important question: **Is the performance improvement due to the aggregation function and loss function introduced by the authors, or simply due to the use of \( H^{(0:l-1)} \)?**

To clarify this, I strongly recommend an **ablation study** where everything remains unchanged except that \( H^{l-1} \) is used instead of \( H^{(0:l-1)} \). This would help isolate the true contribution of \( AGG_B \). Without this analysis, the authors' claim remains inconclusive.

**Essential References Not Discussed:**

N.A.

**Experimental Designs Or Analyses:**

1. The performance improvement introduced by \( AGG_B \) appears to be marginal, with gains of less than 1% on many datasets. Considering the standard deviation, it remains inconclusive whether \( AGG_B \) is genuinely effective.
   - I suggest the authors evaluate \( AGG_B \) on deep GNNs, as DropEdge tends to perform better with increased depth. This would provide a clearer understanding of its impact.

2. The authors make claims regarding heterophilous graphs, yet they only test on two heterophilous datasets, and one of which is problematic.
   - A broader evaluation on more diverse heterophilous datasets is necessary to support these claims.

3. Given the marginal performance improvements and large standard deviations, it would be beneficial to include experiments on larger-scale datasets to assess \( AGG_B \)’s scalability and effectiveness.

Overall, the results presented are not convincing, and additional experiments—especially on deeper GNNs, more heterophilous datasets, and larger-scale benchmarks—are necessary to substantiate the claims.

**Methods And Evaluation Criteria:**

1. Novelty of the method is limited. For example, two stage training with dropedge is proposed in Tuneup [1].
2. the Chameleon dataset used in the paper is known to be problematic[2]. Please use the filtered dataset instead.


[1] Hu, Weihua, et al. "TuneUp: A Simple Improved Training Strategy for Graph Neural Networks." arXiv preprint arXiv:2210.14843 (2022).
[2] Platonov, Oleg, et al. "A critical look at the evaluation of GNNs under heterophily: Are we really making progress?." arXiv preprint arXiv:2302.11640 (2023).

**Other Comments Or Suggestions:**

N.A.

**Other Strengths And Weaknesses:**

The writing and presentation are clear and well-structured.

**Questions For Authors:**

please see weaknesses above.

**Relation To Broader Scientific Literature:**

Two stage training of GNN with the use of Dropedge is proposed in Tuneup. The training framework is similar.

**Theoretical Claims:**

No, i did not check the proofs in the appendix.

---

> ### Author Rebuttal · Authors · 2025-04-01
>
> We sincerely appreciate your thoughtful review. Our response is organized around three points: (1) novelty, (2) ablation study on the AGG$_B$ design, and (3) additional experiments.
>
> **Q1. The novelty of the method is limited. For example, two-stage training with DropEdge is proposed in TUNEUP.**
>
> There were previous works that considered degree bias and structural disparity as separate issues.
> Our main contribution is to offer a new perspective, reframing these issues as instances of a broader problem: **structural inconsistency**.
> We propose AGG$_B$, which directly addresses this general problem.
> To the best of our knowledge, no previous work has solved these two problems at once or even considered these problems together, but our AGG$_B$ consistently and significantly outperforms the approaches designed specifically for certain problem.
>
> TUNEUP performs two-stage training to utilize the pseudo-labels from a classifier for semi-supervised node classification.
> The reason why they used DropEdge is to reduce the effect of overfitting caused by the imperfect knowledge of pseudo labels; our approach is totally different.
> Our two-stage training is motivated by our theoretical results showing that existing GNNs are inherently impossible to solve the structural inconsistency.
> To bypass the inherent limitation of GNNs, we separate robustness optimization into a second stage and introduce a carefully designed parameter block, AGG$_B$, specifically tailored to optimize robustness effectively.
>
> **Q2. Is the performance improvement simply due to the use of $H^{(0:l-1)}$?**
>
> No. In response to your suggestion, we strengthened our ablation study ([link - Table A, B](https://shorturl.at/2cblM)) by including a single-layer  variant, $(D+I)^{-1}H^{l-1}W^{l}$ and extending across all datasets (beyond the 4 originally used).
> This variant satisfies conditions C1 and C2 while limiting the usable information to the immediate previous layer.
> The results show that while alternative layer architectures provide gains under our training scheme and loss, our original AGG$_B$ consistently works the best.
> It is also noteworthy that this experiment has a comparison with JKNet-style block, which also uses the information from all previous layers, and our AGG$_B$ consistently outperforms it.
> Following your concerns, we also replaced Chameleon with its filtered version and added filtered Squirrel. The overall performance trends remain consistent.
>
> In fact, the reason why we chose to use all previous layers is not because of performance.
> If AGG$_B$ fails to fully resolve the structural discrepancies at intermediate layers, these inconsistencies may propagate into deeper layers, making the effect of AGG$_B$ partially reflected.
> By referencing earlier representations, AGG$_B$ can access less corrupted information.
> We will add more in-depth discussion on the choice of our parameter block in the final version.
>
> **Q3. Marginal performance improvement and more additional experiments is requires**
>
> We believe that a difference in experimental setups should be considered when evaluating the significance of improvements.
> Unlike many studies that train GNNs from scratch, our method is applied to trained GNNs.
> Few approaches—such as curriculum learning (e.g., TUNEUP) or graph augmentation (e.g., GraphPatcher)—operate in this setting.
> While prior works often fix hyperparameters for base GNNs, we performed extensive grid searches, demonstrating that our method yields gains beyond what hyperparameter tuning can achieve—by addressing fundamental architectural limitations.
> Although this setup may boost base accuracy and lead to smaller apparent gains, we believe it more accurately reflects the true effectiveness and robustness of our method.
>
> Following your advice, we conducted three additional experiments:
>
> (1) **Performance under edge removal** ([link - Table C](https://shorturl.at/GjlH9)) :
> We directly evaluated robustness driven by AGG$_B$ under random edge removal. AGG$_B$ significantly outperformed standard GCNs—even those trained with DropEdge—demonstrating its edge robustness.
>
> (2) **Experiments with deeper GCNs** ([link - Table D](https://shorturl.at/bhlls)) :
> AGG$_B$ improved performance in 28 out of 30 configurations, with larger gains at greater depths, where increased aggregations makes models more vulnerable to structural inconsistency. Even on GCNs trained with DropEdge, AGG$_B$ boosted performance in 28 out of 30 cases, highlighting its distinct mechanism.
>
> (3) **Experiments on larger datasets** ([link - Table E, F](https://shorturl.at/DvhF3)) :
> On larger datasets, AGG$_B$’s performance remained consistent with earlier observations, underscoring its broad applicability.
>
> We appreciate your valuable insights and important concerns of our work. We will incorporate these additional findings in the final version. Thank you again for your detailed feedback.

---

### Official Review · Reviewer_G3VD · 2025-03-14

**Overall Recommendation:** 4

**Summary:**

This paper analyzes DropEdge, which is widely used in GNNs. it shows that DropEdge has limited effectiveness for GNNs. Through theoretical analysis, the authors show the limitation comes from fundamental constraints in GNN architectures. They propose "Aggregation Buffer" (AGGB), a parameter block that can be added to any pre-trained GNN and trained with DropEdge. AGGB addresses structural inconsistencies in graphs, improving performance on 11 benchmark datasets while effectively mitigating common GNN problems like degree bias and structural disparity.

## After rebuttal

I am satisfied wth the authors' response.

**Claims And Evidence:**

I found the following claims in the paper.
1.  DropEdge has limited performance gains in supervised learning tasks due to fundamental limitations in GNN architectures. This has been well verified with both theoretical analysis and empirical evidence
2.  The limited effectiveness of DropEdge stems from the AGG operation in GNNs and its inability to maintain consistent representations under structural perturbations. The authors develop a theoretical framework using discrepancy bounds (Theorems 3.8 and 3.9) to show that unlike MLPs, GCNs cannot establish a constant discrepancy bound independent of input when adjacency matrices differ.
3. Aggregation Buffer (AGGB) effectively addresses GNN limitations. This is also well supported with both theory and emprical results.
4. AGGB consistently improves performance across different GNN architectures and datasets.  Comprehensive results in experiment study support this claim.

**Essential References Not Discussed:**

I think the related work part is already good. There are no other essential references that should be discussed.

I think it would be better if the author also included some discussions on general data augmentation techniques in graphs, not limited to random drop-based.  For example, the ones based on mixup.

[1] G-Mixup: Graph Data Augmentation for Graph Classification (ICML 22)

Besides, for the drop-based ones, I also suggest adding some references about adative dropping, like

[1] Robust Graph Representation Learning via Neural Sparsification [ICML 20]
[2] Learning to Drop: Robust Graph Neural Network via Topological Denoising (WSDM 21)
[3] xAI-Drop: Don't use what you cannot explain (LOG 24)

**Experimental Designs Or Analyses:**

Yes.   I am convinced with the experimental design. Almost all benchmark datasets with different sizes are used.   DropEdge and other drop based methods like Drop Message are included as baselines.  I am convinced by the experiments.  The ablation studies and in-depth analysis are also comprehensive.

I have a minor concern/suggestion.
The authors mention that "use a 10%/10%/80% split for training, validation, and test, which is common for semi-supervised learning."  I suggest the authors revise this sentence.  Since in the literature, they usually don't use this setting. For example, the cora, they often have two settings, one is with 20 samples for training per class; the other setting is with 1000 for testing, 500 for validation, and the others for training.

**Methods And Evaluation Criteria:**

The proposed methods and evaluation criteria in this paper are well-justified and appropriate for addressing GNN robustness issues. The Aggregation Buffer design directly targets the identified theoretical limitations in GNN architectures with its degree-normalized linear transformation satisfying both edge-awareness and stability conditions. The two-step training approach is new and sound.

For evaluation, the paper employs a comprehensive approach using 11 diverse benchmark datasets, comparing against relevant baselines, The thorough ablation studies examine different AGGB architectures, loss functions, hyperparameter variations, and component contributions.

**Other Comments Or Suggestions:**

I don't have other comments.

**Other Strengths And Weaknesses:**

Strengths
1.Strong theoretical foundation that connects the empirical limitations of DropEdge to fundamental properties of GNN architectures
2. Novel characterization of the bias-robustness trade-off in GNN training that explains previously observed phenomena
3. The proposed AGGB is simple yet effective, requiring minimal computational overhead
4. Extensive empirical validation across diverse datasets demonstrates practical utility

Weaknesses

1. The paper focuses exclusively on node classification tasks; application to other GNN tasks
2. The theoretical analysis primarily focuses on GCNs; stronger theoretical connections to other GNN architectures would strengthen the paper

**Questions For Authors:**

1.  the current paper focuses on node classification tasks, but could the Aggregation Buffer approach be effectively applied to other common GNN tasks such as graph classification and link prediction? Would the theoretical insights about discrepancy bounds and the bias-robustness trade-off transfer directly to these tasks, or would they require significant adaptation?
2. the theoretical analysis primarily focuses on GCNs,  to what extent can your findings regarding discrepancy bounds be generalized to other GNN architectures like GAT, GraphSAGE, or GIN?

**Relation To Broader Scientific Literature:**

This paper's work on improving GNN robustness through Aggregation Buffer connects to several research threads in graph learning. It extends theoretical understanding of DropEdge by providing a formal analysis of its limitations, building on discrepancy bound concepts commonly used in domain adaptation. The paper's findings on GNN vulnerability to structural perturbations align with literature on GNN over-smoothing. Its two-stage training approach with parameter freezing shares conceptual similarities with knowledge distillation techniques. By framing various structural inconsistency problems (degree bias, heterophily) as manifestations of edge-robustness limitations, the paper offers a unifying perspective that connects previously separate research directions in GNN architecture design.

**Theoretical Claims:**

Yes, I briefly check the theorical claims, but not in detail. I don't find errors.

---

> ### Author Rebuttal · Authors · 2025-04-01
>
> We sincerely appreciate your detailed review and insightful questions. We hope the responses below address your concerns.
>
> **Revising the sentence explaining the way of data split**
>
> Thank you for your thoughtful suggestion. We acknowledge that for datasets such as Cora, Citeseer, and Pubmed, the settings you mentioned are more commonly used.
> In the final version, we will revise it to:
> ``Experiments are repeated 10 times, each with a independently randomized 10\%/10\%/80\% split for training, validation, and test, respectively."
> We hope this clarification addresses your concern.
>
> **Discussions related to other data augmentation techniques in graph, non-dropping(e.g G-Mixup) and adaptive dropping(e.g NeuralSparse and xAI-Drop) methods.**
>
> G-Mixup is inspired by Mixup in computer vision and generates synthetic graphs by interpolating between estimated graphons of different groups. While effective for graph classification, it is not directly applicable to node-level tasks—our focus—since node classification typically involves a single large graph rather than many samples.
> NeuralSparse trains a dedicated sparsification network to remove task-irrelevant (noisy) edges deterministically. In this sense, it functions more as an architectural variant preceding the GNN than as a stochastic augmentation method.
>
> xAI-Drop drops nodes with the low fidelity scores in high probability and can be viewed as a parametric, biased variant of DropNode. This method also produces reduced subgraphs (Definition 3.2), and such adaptive strategies could replace DropEdge in our framework to yield richer structural signals and potentially boost AGG$_B$’s performance.
> We view this integration as a promising direction for future work and will include this discussion in the final version.
>
> **W1. Could the Aggregation Buffer approach be effectively applied to other common GNN tasks such as graph classification and link prediction?**
>
> For graph classification, the readout function aggregates node representations into a graph-level output.
> We can view the input as a set of rooted subgraphs, $S\_i ={\\{G_j\\}}^{m}\_{j=1}$, where $m$ is the number of nodes in graph $i$.
> The classification objective can then be expressed as $D_{KL}(P(y_i|S_i)||Q(y_i|S_i))$.
> Since DropEdge produces a set of reduced subgraphs $\tilde{S}_i$, our bias–robustness framework can be applied similarly.
>
> For link prediction, framed as a binary classification task for edge existence, the model input can be considered as a pair of rooted subgraphs, $\pi_{(u, v)} =\{G_u, G_v\}$.
> The objective becomes $D_{KL}(P(y_{(u,v)}|\pi_{(u,v)})||Q(y_{(u,v)}|\pi_{(u,v)}))$, with data augmentation defined analogously on, $\tilde{\pi}_{(u,v)}$.
>
> Although these tasks share the bias–robustness trade-off perspective, each requires task-specific adaptations. In graph classification, the set of node representations at each intermediate layer varies with node dropping, making discrepancy between different-sized sets hard to define. In link prediction, operations such as dot product, absolute difference, Hadamard product, or concatenation followed by an MLP require separate theoretical treatment. Thus, while the bias–robustness trade-off view can be extended, AGG$_B$’s layer-wise correction mechanism demands new theoretical insights and task-specific methods—a promising direction for future work.
>
> **W2. To what extent can your findings regarding discrepancy bounds be generalized to other GNN architectures?**
>
> Thank you for this question. We extended our discrepancy analysis beyond GCNs by considering the more generalized layer-wise update:
> $$
> \mathbf{H}^{(l)} = \sigma(\mathbf{A}^* \mathbf{H}^{(l-1)} \mathbf{W}_1^{(l)} + c \mathbf{H}^{(l-1)} \mathbf{W}_2^{(l)}),
> $$
> where $A^*$ is a (possibly normalized) adjacency matrix and $c$ is a constant. Under this formulation, Theorem 3.9 still holds, with constants $C_1$ and $C_2$ depending on model parameters and adjacency differences, can be shown by extending the arguments in our proof at Appendix D.
> This abstraction encompasses a broader GNN architectures. For instance, inductive GCN corresponds to $A^* = (D+I)^{-1}A$ and $c=0$; GraphSAGE uses $A^* = D^{-1}A$ and $c=1$; GIN sets $A^* = A$ and $c = 1 + \epsilon^{(l)}$; and for GAT, with $c=0$ and we can bound the norm of $A^*$ due to its row-stochasticity.
> Thus, our theoretical findings regarding discrepancy bounds apply to a wide range of GNN architectures. A formal proof for these cases will be included in the final version.
>
> Thank you again for your insightful review and for taking the time to consider our responses.

---

### Decision · Program_Chairs · 2025-05-01

**Decision:**

Accept (poster)

**Comment:**

This paper proposes Aggregation Buffer (AGGB), a simple yet effective parameter block designed to enhance GNN robustness by addressing the limitations of DropEdge. The approach is backed by solid theoretical analysis using discrepancy bounds and is compatible with various GNN architectures.
The authors provide comprehensive experiments across multiple datasets and architectures, including deeper and larger models, and address reviewer concerns thoroughly in the rebuttal. While some reviewers noted that performance gains are modest, the conceptual novelty and practical utility of AGGB make the contribution meaningful.
Reviewer jKJV has questioned the novelty of the proposed AGGB relative to TuneUp, yet, the authors have convincingly highlighted the differences in the approach and particularly the purpose.

Final recommendation: Accept (poster). The paper offers a well-motivated, general solution to structural inconsistencies in GNNs and is of value to the ICML community.